# Reconstructive Visual Instruction Tuning

**Haochen Wang**[1,2]    **Anlin Zheng**[3]    **Yucheng Zhao**[4†]    **Tiancai Wang**[4*]
**Zheng Ge**[5]    **Xiangyu Zhang**[4,5]    **Zhaoxiang Zhang**[1,2*]

[1] New Laboratory of Pattern Recognition,
  State Key Laboratory of Multimodal Artificial Intelligence Systems,
  Institute of Automation, Chinese Academy of Sciences
[2] University of Chinese Academy of Sciences
[3] University of Hong Kong    [4] MEGVII Technology    [5] StepFun
  {wanghaochen2022, zhaoxiang.zhang}@ia.ac.cn    wangtiancai@megvii.com

Project Page: https://haochen-wang409.github.io/ross

## Abstract

This paper introduces **r**eco**n**structive vi**s**ual in**s**truction tuning (**Ross**), a family of Large Multimodal Models (LMMs) that exploit *vision-centric supervision signals*. In contrast to conventional visual instruction tuning approaches that exclusively supervise text outputs, **Ross** prompts LMMs to supervise *visual outputs* via reconstructing input images. By doing so, it capitalizes on the inherent richness and detail present within input images themselves, which are often lost in pure text supervision. However, producing meaningful feedback from natural images is challenging due to the heavy spatial redundancy of visual signals. To address this issue, **Ross** employs a denoising objective to reconstruct latent representations of input images, avoiding directly regressing exact raw RGB values. This *intrinsic activation* design inherently encourages LMMs to maintain image detail, thereby enhancing their fine-grained comprehension capabilities and reducing hallucinations. Empirically, **Ross** *consistently* brings significant improvements across different visual encoders and language models. In comparison with *extrinsic assistance* state-of-the-art alternatives that aggregate multiple visual experts, **Ross** delivers competitive performance with a single SigLIP visual encoder, demonstrating the efficacy of our vision-centric *supervision* tailored for *visual outputs*.

## 1 Introduction

The success of GPT-style Large Language Models (LLMs) (Radford et al., 2018; 2019; Brown et al., 2020; OpenAI, 2023b; Yang et al., 2024a; Touvron et al., 2023; Chiang et al., 2023; Dubey et al., 2024) has motivated researchers to adapt LLMs to understand multimodal inputs (Liu et al., 2023a; 2024a; Dai et al., 2023; Bai et al., 2023). Notably, visual instruction tuning approaches (Liu et al., 2023a) demonstrate superior performance with cost-efficient training recipes. Some approaches (Chen et al., 2024b; Li et al., 2024c) even surpass GPT-4V(ision) (OpenAI, 2023a) on benchmark evaluations.

Typically, these Large Multimodal Models (LMMs) based on visual instruction tuning adopt a plug-in architecture, as depicted in Figure 1a, where pre-trained vision-language foundation models such as CLIP (Radford et al., 2021) are responsible for projecting images into visual tokens. They serve as prefix tokens for multimodal comprehension. However, this type of design, *i.e.*, *visual encoder → connector → LLM ⇐ language instructions*, where "⇐" indicates supervision, is primarily LLM-centric: *(i)* visual comprehension largely depends on vision-to-text alignment and the selected vision models, and *(ii)* *supervision* derives exclusively from text data. As a result, they exhibit systematic visual shortcomings such as recognizing specific visual patterns (Tong et al., 2024b).

Until very recently, some concurrent works proposed *vision-centric* solutions (Tong et al., 2024a;b). Illustrated in Figure 1b, their solutions leverage *extrinsic assistance* via aggregating several different visual experts. Inspired by the evolution in image recognition, from manually designed visual

---

*Corresponding authors. † Project lead.

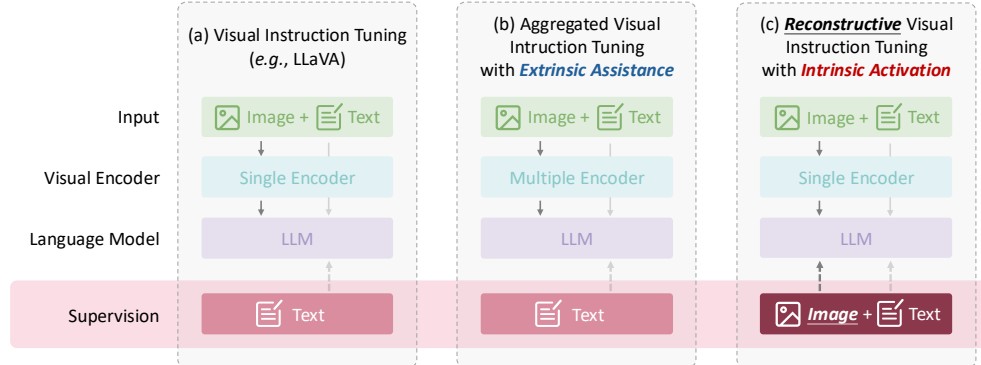

Figure 1: **Conceptual comparison** between different pipelines. **(a)** Typical visual instruction tuning approaches (Liu et al., 2023a; 2024a) follow a LLM-centric design that solely leverage text supervision. **(b)** Aggregated visual instruction tuning alternatives (Tong et al., 2024a;b) leverages *extrinsic assistance* via combining several visual experts, requiring a careful selection of visual experts. **(c)** Our ROSS, with a single visual encoder, *e.g.*, CLIP (Radford et al., 2021) and SigLIP (Zhai et al., 2023), designs extra vision-centric reconstructive supervision as *intrinsic activation*. In this way, LMMs are required to preserve every detail of input images, thereby enhancing multimodal comprehension capabilities and reducing hallucinations.

features (Sánchez & Perronnin, 2011) to learnable deep convolutional models (Krizhevsky et al., 2012), we suggest that *intrinsic activation* offers a more viable path forward. Just as deep models automatically learn hierarchical and abstract features from raw data, we believe *intrinsic activation* methods are similarly more adaptable for multimodal comprehension, reducing reliance on hand-crafted engineering, thereby enhancing both generalization and performance. Therefore, we aim to explore *intrinsic activation* solutions based on the following principles:

1. **Supervise Visual Outputs.** Current LMMs solely supervise text outputs, neglecting a significant amount of visual outputs *unused*. For instance, LLaVA-v1.5 (Liu et al., 2024a) utilizes 576 visual tokens to represent a single $336 \times 336$ image, yet their corresponding outputs remain *unsupervised*. Intuitively, since input images themselves inherently provide rich and detailed information, we regard LMMs *reconstructing input images* as the supervision of those visual outputs. This approach encourages LMMs to maintain low-level details, thereby enhancing their fine-grained comprehension abilities and reducing hallucinations.

2. **Explore the Optimal Formulation.** Designing this *self-supervised* task effectively is not straightforward. Motivated by the success of *masked* autoencoder (He et al., 2022) compared to its basic version denoising autoencoder (Vincent et al., 2008), we identify handling *heavy spatial redundancy of visual signals* as the underlying key factor. To this end, we formulate our approach as follows: **(i)** for reconstruction *targets*, instead of raw RGB pixels, we make LMMs reconstruct *latent visual tokens*, and **(ii)** for reconstruction *objectives*, to avoid directly regressing exact token values, we adopt per-token *denoising*.

To this end, we propose ROSS, termed of **r**econstructive **v**isual **i**nstruction tuning, which utilizes input images as direct supervision signals illustrated in Figure 1c. Technically, to address the spatial redundancy inherent in natural visual signals (He et al., 2022), we train a small denoising network, which takes high-level visual outputs $x$ as conditions to recover low-level fine-grained visual tokens $z$, representing an underlying distribution $p(z|x)$. These latent tokens $z$ are derived from a frozen teacher tokenizer such as continuous VAE (Kingma, 2013) and discrete VQGAN (Esser et al., 2021). Unlike *extrinsic assistance* solutions (Tong et al., 2024a;b), our *intrinsic activation* solution naturally maintains a lightweight inference procedure. More importantly, when adapting to new visual domains, our solution avoids a careful choice of new domain-specific experts, *e.g.*, MiDaS-3.0 (Birkl et al., 2023) for understanding depth maps, which is more efficient and easier to implement.

Empirically, ROSS achieves top performance across a wide range of multimodal comprehension benchmarks. Notably, our ROSS excels in fine-grained vision-centric benchmarks (Tong et al., 2024b; Masry et al., 2022) and hallucination benchmarks (Guan et al., 2024; Li et al., 2023c). To be specific, with a *single* SigLIP (Zhai et al., 2023) as the visual encoder, **ROSS-7B** achieves 57.3 on

HallusionBench (Guan et al., 2024) and 54.7 on MMVP (Tong et al., 2024b), significantly outperforms state-of-the-art alternatives with similar model sizes which aggregate several visual experts as *extrinsic assistance*, *e.g.*, Cambrian-1-8B (Tong et al., 2024a). In-depth analysis demonstrates the effectiveness of **ROSS** for directing focus towards visual elements and understanding depth maps. We hope our research will inspire future work in designing supervision signals for large multimodal models.

## 2 RELATED WORK

**Visual Instruction Tuning.** Most visual instruction tuning-based LMMs adopt a plug-in architecture (Liu et al., 2023a; 2024a; Bai et al., 2023; Wang et al., 2024c), where a language-supervised visual encoder (Radford et al., 2021; Zhai et al., 2023) is responsible for extracting visual tokens. A connector is used to map those visual representations into the LLM space, *e.g.*, Resamplers (Alayrac et al., 2022), Q-Formers (Li et al., 2023b; Dai et al., 2023; Bai et al., 2023; Ge et al., 2024a), and MLPs (Liu et al., 2023a; 2024a; Li et al., 2024c; Liu et al., 2024b; Li et al., 2024a). These LMMs usually follow a two-stage training recipe. During the alignment stage, the connector is trained on high-quality caption data. Next, the full model is trained on single-image visual instruction tuning data. However, *only text outputs* are supervised. **ROSS**, on the other hand, introduces novel vision-centric supervision via reconstructing fine-grained visual tokens conditioned on *visual outputs*.

**Visual Encoders for LMMs.** As the original CLIP (Radford et al., 2021) adopted by conventional visual instruction tuning approaches is trained on noisy image-text pairs, it exhibits specific visual shortcomings, and thus stronger backbones (Fang et al., 2024; Zhai et al., 2023; Chen et al., 2024c) have been introduced to LMMs. Some concurrent works (Tong et al., 2024b;a) leverage *extrinsic assistance*, which further utilizes vision-only self-supervised models (Oquab et al., 2023; Wang et al., 2023a;b;c; He et al., 2022; Caron et al., 2021) and domain experts (Kirillov et al., 2023; Birkl et al., 2023; Shen et al., 2023; Gu et al., 2024; Zhang et al., 2024a; Wang et al., 2022; 2024a;b). **ROSS**, from a new *intrinsic activation* perspective, aims to catalyze enhanced comprehension through *reconstructing input images* with *no* extra visual experts.

**Generative Objectives for LMMs.** Another line of work introduces *pre-trained* text-to-image diffusion models (Rombach et al., 2022) to make LMMs capable of both comprehension and *generation* (Dong et al., 2024; Ge et al., 2024a; Sun et al., 2024b; Ge et al., 2024b; Sun et al., 2023; Ren et al., 2025). Our **ROSS**, with a totally different motivation, targets to catalyze multimodal comprehension via *reconstruction*. Specifically, conditions are different, where Dong et al. (2024) and Sun et al. (2024b) take outputs corresponding to *learnable queries* as conditions, while our **ROSS** takes outputs corresponding to *visual inputs*. Those methods are *generative* while **ROSS** is *reconstructive*. The detailed pipeline comparison can be found in Appendix C.

## 3 PRELIMINARIES

**Large Multimodal Models.** In the literature (Radford et al., 2018; 2019), a $\theta$-parameterized LLM models the canonical *causal* distribution of each *text* token $\boldsymbol{x}_i$ as $p_\theta(\boldsymbol{x}) = \prod_{i=1}^{T} p_\theta(\boldsymbol{x}_i|\boldsymbol{x}_{<i})$, where $\{\boldsymbol{x}_i\}_{i=1}^{T}$ represents a sequence of text tokens. To make LLMs understand visual contents, typical plug-in style LMMs (Liu et al., 2023a; 2024a) regard a sequence of visual tokens as prefix tokens. Specifically, an input image $\boldsymbol{I} \in \mathbb{R}^{H \times W \times 3}$ is first projected into a sequence of visual tokens by a $\xi$-parameterized visual encoder $\mathcal{G}_\xi$ such as CLIP (Radford et al., 2021) and SigLIP (Zhai et al., 2023), where $(H, W)$ indicates the spatial resolution. Then, a $\phi$-parameterized multimodal projector $\mathcal{H}_\phi$ is utilized to project these visual tokens into the feature space of LLMs. As a result, the canonical causal distribution in a *multimodal* sentence containing an image $\boldsymbol{I}$ becomes

$$p_\Theta(\boldsymbol{x}) = \prod_{i=1}^{T} p_\Theta(\boldsymbol{x}_i|\boldsymbol{x}_{<i}, \boldsymbol{v}), \quad \boldsymbol{v} = \mathcal{H}_\phi \circ \mathcal{G}_\xi(\boldsymbol{I}), \tag{1}$$

where $\Theta = \{\theta, \xi, \phi\}$ is the parameters and $\boldsymbol{v} \in \mathbb{R}^{N \times D}$ indicates the projected visual tokens. $N$ is the number of visual tokens and $D$ indicates the feature channel. The visual encoder $\mathcal{G}_\xi$ could be either frozen (Liu et al., 2023a; 2024a; Tong et al., 2024a) or fine-tuned (Liu et al., 2024b; Bai et al., 2023; Li et al., 2024c; Wang et al., 2024e).

**Training Recipes for LMMs.** LMMs almost follow a two-stage training recipe (Liu et al., 2023a), *i.e.*, the pre-training stage (or the alignment stage) and the supervised fine-tuning stage (or the

instruction tuning stage). The instruction (supervision) comes from languages such as the answers to VQA tasks, maximizing the log-likelihood of *text* outputs:

$$\mathcal{L}_{\text{LMM}}^{\text{text}}(\Theta = \{\theta, \xi, \phi\}, \boldsymbol{x}, \boldsymbol{I}) = \frac{-1}{T-N} \sum_{i=N+1}^{T} \log p_{\Theta}(\boldsymbol{x}_i | \boldsymbol{x}_{<i}, \boldsymbol{v}), \tag{2}$$

where $N$ represents the number of visual tokens and visual outputs (one input token corresponds to one visual output). From Equation (2), we can tell that only text outputs $\boldsymbol{x}_{i>N}$ are supervised.

## 4  ROSS: RECONSTRUCTIVE VISUAL INSTRUCTION TUNING

In this section, we first provide an overview of our reconstructive visual instruction tuning (**ROSS**). Then, we discuss our explorations towards the optimal formulation in the following subsections, with the ultimate goal of *handling spatial redundancy of visual signals* to provide meaningful visual supervision. Our explorations mainly include reconstruction *targets* and the training *objective*.

**Overview.** Illustrated in Figure 2, the overall philosophy of our **ROSS** is to construct *reconstructive* visual supervision signals on visual outputs $\boldsymbol{x}_{i\leq N}$. The training objective includes *(i)* the original next-token prediction on $\boldsymbol{x}_{i>N}$ shown in the right part of Figure 2, and *(ii)* another *reconstructive term* in the *left* part of Figure 2, *i.e.*, $\mathcal{L}_{\text{Ross}} = \mathcal{L}_{\text{LMM}}^{\text{text}} + \mathcal{L}_{\text{LMM}}^{\text{visual}}$. Specifically, this visual term could be any custom measurements $\mathcal{M}$ between $\boldsymbol{x}_{i\leq N}$ and specific reconstruction targets of image $\boldsymbol{I}$:

$$\begin{aligned} \mathcal{L}_{\text{LMM}}^{\text{visual}}(\Theta &= \{\theta, \xi, \phi, \pi\}, \boldsymbol{x}, \boldsymbol{I}) \\ &= \mathcal{M}(\mathcal{J}_{\pi}(\boldsymbol{x}_{i\leq N}), \mathcal{F}(\boldsymbol{I})), \end{aligned} \tag{3}$$

where $\mathcal{J}_{\pi}$ indicates the $\pi$-parameterized post projection that maps the dimensions of visual tokens $\boldsymbol{x}_{i\leq N}$ to be consistent with the teacher tokenizer $\mathcal{F}$.

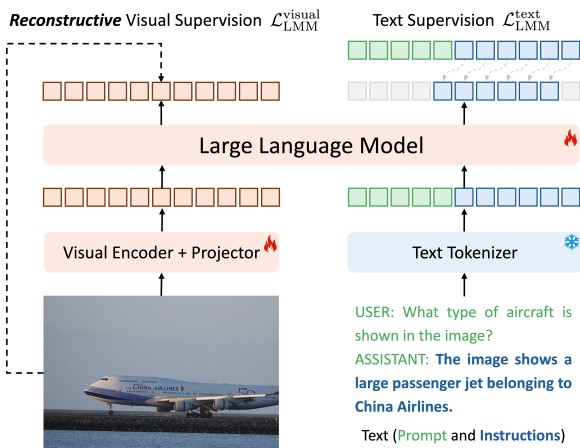

Figure 2: **Overview of ROSS.** Given an input image and the corresponding text to this image, **ROSS** aims to *supervise visual outputs by reconstruction.*

**Variants of ROSS.** Evidently, different choices of $\mathcal{F}$ and $\mathcal{M}$ contribute to different variants. $\mathcal{F}$ controls the reconstruction *target* while $\mathcal{M}$ defines the *objective*:

1. Towards the *target*, $\mathcal{F}$ can be the pachify operation (Dosovitskiy et al., 2021), resulting in *pixel-level* reconstruction, or pre-trained fine-grained visual tokenizers such as VAE (Kingma, 2013) and VQGAN (Esser et al., 2021), leading to *latent-level* reconstruction. $\mathcal{F}$ could even be vision-only models such as DINOv2 (Oquab et al., 2023), making LMMs learn specific visual patterns from $\mathcal{F}$, which is also a type of *latent-level* reconstruction.

2. Towards the *objective*, the most straightforward choice of $\mathcal{M}$ is MSE or cosine similarity for *regressing* raw pixel values or latent features, respectively. We also explore the *denoising* objective (Ho et al., 2020) to avoid being overwhelmed by fitting exact values.

We introduce our explorations step by step in the following sections. The ultimate goal of our exploration is to design an appropriate self-supervised reconstructive pre-text task that provides meaningful vision-centric supervision signals to LMMs, where handling the *spatial redundancy* of visual signals (He et al., 2022) becomes the crux.

### 4.1  ROSS$^{\text{R}}$: REGRESSING AS RECONSTRUCTIVE VISUAL INSTRUCTION

In this section, we introduce straightforward variants, *i.e.*, *regressing* as reconstructive visual instruction. As shown in Figure 3, depending on the choice of $\mathcal{F}$, it mainly has three variants: (a) **ROSS$^{\text{R}}$**-Pixel, (b) **ROSS$^{\text{R}}$**-Latent, and (c) **ROSS$^{\text{R}}$**-Latent2Pixel.

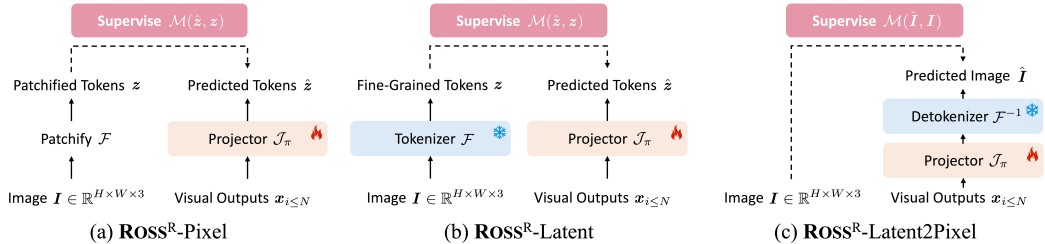

Figure 3: Variants of **ROSS**$^R$, where *regression* objectives are either computed on raw RGB values in (a) and (c), or specific latent space determined by $\mathcal{F}$ in (b). We adopt MSE as $\mathcal{M}$ for *pixel* regression in (a) and (c), and cosine-similarity for *latent* regression in (b), respectively.

**Directly Regressing Raw RGB Values.** The most straightforward variant is to directly regress raw RGB values illustrated in Figure 3a, called "**ROSS**$^R$-Pixel". Under such a setting, $\mathcal{F}$ is the patchify operation (Dosovitskiy et al., 2021), reshaping the image $I \in \mathbb{R}^{H \times W \times 3}$ into a sequence of flattened 2D patches $I_p \in \mathbb{R}^{N \times (3P^2)}$, where $(P, P)$ is the resolution of each image patch and $N = HW/P^2$ indicates the resulting number of patches. $\mathcal{J}_\pi$ can be a simple MLP, mapping the dimension of visual outputs $x_{i \leq N}$ from $D$ to $3P^2$. The measurement $\mathcal{M}$ is MSE. However, as visual signals suffer from *heavy spatial redundancy* (He et al., 2022), such a design may not provide meaningful supervision to LMMs. An intuitive alternative to avoid directly regressing raw RGB values while still reconstructing the image is to urge LMMs to reconstruct *latent tokens*, introduced as follows.

**Regressing Latent Tokens.** Illustrated in Figure 3b, **ROSS**$^R$-Latent aims to regress fine-grained *latent* tokens extracted by the teacher tokenizer. $\mathcal{F}$ can be models trained with discriminative tasks such as DINOv2 (Oquab et al., 2023) and DEIT-III (Touvron et al., 2022). The *encoder* part of models trained with reconstruction tasks such as VQGAN (Esser et al., 2021) and VAE (Kingma, 2013) are also capable. $\mathcal{M}$ here is the consine-similarity. Intuitively, the *decoder* part of the latter is able to remap latent tokens into the pixel space. Therefore, supervising in the pixel space via *decoding* becomes another valid variant introduced as follows.

**Regressing RGB Values via Decoding.** Shown in Figure 3c, **ROSS**$^R$-Latent2Pixel requires a *decoder* to project predicted latent tokens $\hat{z}$ into the RGB pixel space, resulting in predicted image $\hat{I}$. Let $\mathcal{F}^{-1}$ be the *decoder* part of VQGAN (Esser et al., 2021) or VAE (Kingma, 2013), and the *regressive* MSE objective $\mathcal{M}$ is performed on pixel-space. Note that we simply use $\mathcal{F}^{-1}$ to represent the decoding process, which is actually *not* the inverse function of $\mathcal{F}$ mathematically.

**Discussion.** Recall that we need to find the optimal solution to address the spatial redundancy of natural visual signals, the *target-level* exploration above achieves this goal *partially*, as the *objective* is limited to vanilla regression. To this end, inspired by Ho et al. (2020) and Li et al. (2024e), we further incorporate a novel *denoising* objective in the following section.

## 4.2 ROSS$^D$: DENOISING AS RECONSTRUCTIVE VISUAL INSTRUCTION

As an objective for handling *heavy spatial redundancy* to provide meaningful vision-centric super-vision signals, denoising is better than vanilla regressing, since the introduction of noise into the training data acts as an implicit form of data augmentation and regularization. The denoising process encourages the model to focus on the underlying data manifold rather than memorizing specific instance values (Chen et al., 2023c; Song & Ermon, 2019; Karras et al., 2022; Yang et al., 2024b).

Techinically, as illustrated in Figure 4a, our final **ROSS**$^D$ takes high-level visual outputs $x_{i \leq N}$ as conditions to recover *clean* fine-grained tokens $z_0$ from *noisy* tokens $z_t$. Specifically, clean tokens $z_0 = \mathcal{F}(I)$ are obtained from the teacher tokenizer $\mathcal{F}$. By default, we utilize a continuous VAE (Kingma, 2013) regularized by Kullback–Leibler (KL) divergence provided by Rombach et al. (2022), since it is believed to capture sufficient image details. The training procedure of the denoiser $\mathcal{J}_\pi$ follows a diffusion process (Ho et al., 2020):

$$\mathcal{L}_{\text{LMM}}^{\text{visual}}(\Theta = \{\theta, \xi, \phi, \pi\}, x, I) = \mathbb{E}_{t,\epsilon}\left[||\mathcal{J}_\pi(z_t; x_{i \leq N}, t) - \epsilon||^2\right]. \quad (4)$$

The denoiser $\mathcal{J}_\pi$ actually estimates the conditional expectation $\mathbb{E}[\epsilon \sim \mathcal{N}(0, I)|z_t]$. More details about the background knowledge of diffusion models can be found in Appendix A.

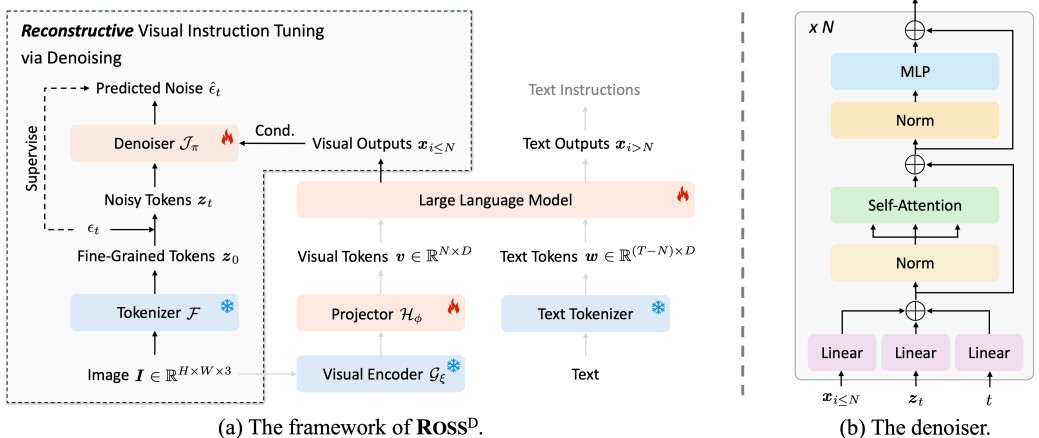

(a) The framework of **ROSS**[D].

(b) The denoiser.

Figure 4: Illustration of (a) the training procedure of **ROSS**[D] and (b) the detailed architecture of the denoiser $\mathcal{J}_\pi$. (a) **ROSS**[D] introduces visual guidance via *denoising fine-grained visual tokens* $z_0$ *conditioning on visual outputs* $x_{i\leq N}$. (b) The denoiser takes noisy tokens $z_t$, current timesteps $t$, and conditions $x_{i\leq N}$ as inputs and outputs the predicted noise $\hat\epsilon_t$. Each denoiser block consists of three linear projection layers and a standard self-attention block (Vaswani et al., 2017).

**Architecture of the Denoiser.** As conditions $x_{i\leq N}$ are *causal*, we introduce a self-attention module to model the inter-token dependencies illustrated in Figure 4b. Specifically, the architecture of the denoiser $\mathcal{J}_\pi$ is a stack of Transformer Encoder blocks (Vaswani et al., 2017) and each block contains three extra projections for conditions $x_{i\leq N}$, inputs $z_t$, and timesteps $t$, respectively.

**Choices of the Teacher Tokenizer.** By default, we adopt *latent* denoising and we take a continuous tokenizer provided by Rombach et al. (2022) as $\mathcal{F}$, since it manages to reconstruct input images with a low rFID (Heusel et al., 2017) and thus it is expected to preserve many low-level details of input images. This extra reconstructive objective, however, is *not* limited to any certain tokenizer $\mathcal{F}$. Discrete tokenizers such as VQGAN (Esser et al., 2021), and vision self-supervised models such as DINOv2 (Oquab et al., 2023), are also qualified to be the tokenizer. Even the patchify operation (Dosovitskiy et al., 2021) is capable, resulting in *pixel* denoising.

## 5 EXPERIMENTS

### 5.1 ABLATION STUDY

**Implementation Details.** All ablation studies are implemented based on LLaVA-v1.5 (Liu et al., 2024a). The visual encoder $\mathcal{G}_\xi$ is CLIP-ViT-L/14@336 (Radford et al., 2021) and the base LLM is Qwen2-7B-Instruct (Yang et al., 2024a). The training data is LLaVA-558K (Liu et al., 2023a) and Cambrian-737K (Tong et al., 2024a) for the pre-training stage and the instruction tuning stage, respectively. We evaluate our each variant of **ROSS** mainly on *(i)* hallucination: POPE (Li et al., 2023c) and HallusionBench (Guan et al., 2024), *(ii)* fine-grained comprehension: MMVP (Tong et al., 2024b) and ChartQA (Masry et al., 2022), and *(iii)* general comprehension: MMBench (Liu et al., 2023b) English dev split. All evaluations are conducted with VLMEvalKit (Duan et al., 2024). Evaluation prompts can be found in Appendix B.

**Pixel Regression *v.s.* Latent Regression.** Starting from the visual instruction tuning baseline (Liu et al., 2023a; 2024a), we first explore the effectiveness of using *regression* as the objective for our reconstructive visual instruction tuning. We utilize a continuous VAE (Kingma, 2013) with an encoder-decoder architecture provided by Rombach et al. (2022), where the *encoder* part serves as $\mathcal{F}$ for **ROSS**[R]-Latent while the *decoder* part is $\mathcal{F}^{-1}$ for **ROSS**[R]-Latent2Pixel. As illustrated in Figure 5, our vision-centric regression supervision outperforms the visual instruction tuning baseline in most cases. Moreover, latent regression performs the best since *regressing raw RGB pixels fails to provide meaningful supervision signals*, regardless of whether utilizing a decoder or not.

**Choices of $\mathcal{F}$.** We study the effectiveness across different latent teacher tokenizers $\mathcal{F}$ in Figure 6, including KL-16 provided by Rombach et al. (2022), which is a continuous VAE (Kingma, 2013) with Kullback–Leibler (KL) divergence, self-supervised DINOv2 (Oquab et al., 2023), fully-supervised

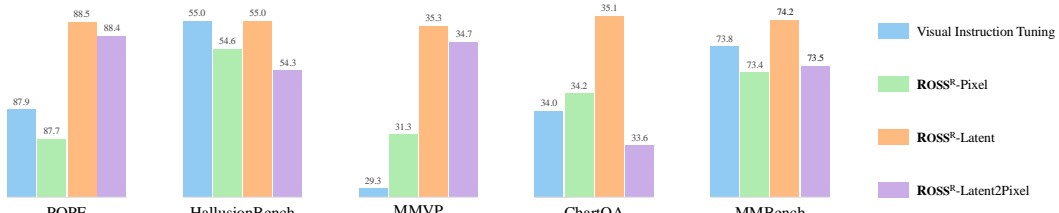

Figure 5: **Pixel Regression *v.s.* Latent Regression.** The teacher tokenizer $\mathcal{F}$ for **ROSS**[R]-Latent is the *encoder* of a continuous VAE (Kingma, 2013) provided by Rombach et al. (2022), while its *decoder* serves as $\mathcal{F}^{-1}$ for **ROSS**[R]-Latent2Pixel. Our vision-centric reconstructive supervision surpasses the visual instruction tuning baseline in most cases. Among three regression variants, **ROSS**[R]-Latent performs the best, as it avoids explicitly regressing redundant raw RGB values.

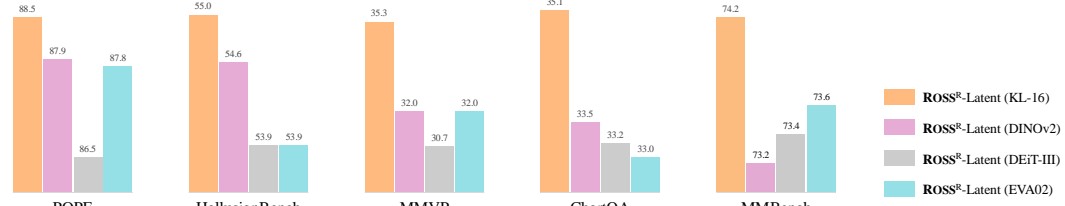

Figure 6: **Choices of the latent teacher tokenizer $\mathcal{F}$.** KL-16 (Rombach et al., 2022) is the best tokenizer as it is originally used for *reconstruction*, and it is expected to preserve the most image details. Other alternatives are utilized for classification (Touvron et al., 2022), instance-level representation learning (Oquab et al., 2023), and language alignment (Fang et al., 2024), respectively.

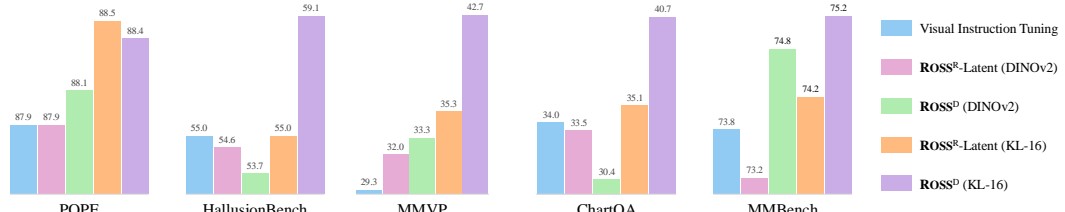

Figure 7: **Regression *v.s.* Denoising.** With KL-16 as the tokenizer, the denoising objective introduced in Equation (4) brings significant improvements over vanilla regression using MSE as it avoids overfitting exact latent token values, even if **ROSS**[R]-Latent (KL-16) has already outperformed the visual instruction tuning baseline by a large margin.

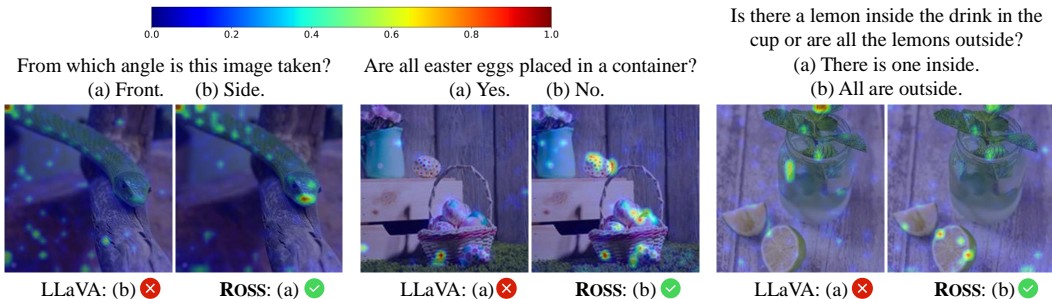

Figure 8: **Qualitative comparison on attention maps** on MMVP (Tong et al., 2024b), where we keep the *same* LLM and training data. With extra vision-centric supervision signals, **ROSS** urges the model to *focus on specific image contents corresponding to the question* with higher attention values.

DEiT-III (Touvron et al., 2022), and language-supervised EVA02CLIP (Fang et al., 2024). Among them, KL-16 is the best choice. One intuitive explanation is that it is expected to preserve the most image details, since it was originally designed to accurately reconstruct input images.

**Regression *v.s.* Denoising.** In Figure 7, we study the effectiveness of the *denoising* objective over vanilla regression across different tokenizers, *i.e.*, KL-16 (Rombach et al., 2022) and DINOv2 (Oquab

Table 2: **Generative *v.s.* Reconstructive.** Following Sun et al. (2024b) and Dong et al. (2024), we adopt 576 learnable latent tokens to *query* the LMM and utilize the corresponding outputs as conditions to the denoiser for generative cases. Extra 102K caption data from ShareGPT4V (Chen et al., 2023b) is introduced to the original SFT data, facilitating text-to-image creation. Reconstructive objectives boost comprehension while generative alternatives *cannot*.

| Method | SFT Data | w/ $\mathcal{L}_{\mathrm{LMM}}^{\mathrm{visual}}$ | | Hallucination | | Fine-Grained | | General |
|--------|----------|------|------|------|-------|------|---------|------|
| | | 737K | 102K | POPE | Hallu. | MMVP | ChartQA | MMB$^{\mathrm{EN}}$ |
| Baseline | 737K + *Caption* 102K | – | – | 86.2 | 55.1 | 32.0 | 30.9 | 74.4 |
| Reconstructive | 737K + *Caption* 102K | ✓ | ✓ | **87.6** | **58.0** | **38.7** | **40.4** | **75.2** |
| Reconstructive | 737K + *Caption* 102K | – | ✓ | 87.6 | 56.3 | 37.3 | 39.7 | 74.9 |
| Generative | 737K + *Creation* 102K | – | ✓ | 85.4 | 52.0 | 30.0 | 31.2 | 73.9 |

Table 3: **The effectiveness of the vision-centric supervision among various LLMs and visual encoders**, where $\mathcal{L}_{\mathrm{LMM}}^{\mathrm{visual}}$ manages to bring significant improvements *consistently*.

| Language Model | $\mathcal{L}_{\mathrm{LMM}}^{\mathrm{visual}}$ | POPE | Hallu. | MMVP | ChartQA | OCRBench | MMB$^{\mathrm{EN}}$ |
|----------------|------|------|--------|------|---------|----------|------|
| *Visual Encoder: CLIP-ViT-L/14@336* | | | | | | | |
| Vicuna-7B-v1.5 | – | 86.3 | 52.5 | 28.0 | 32.9 | 339 | 67.0 |
| | ✓ | **87.2** ↑ 0.9 | **55.8** ↑ 3.3 | **36.0** ↑ 8.0 | **39.8** ↑ 6.9 | **350** ↑ 11 | **67.6** ↑ 0.6 |
| Qwen2-7B-Instruct | – | 87.9 | 55.0 | 29.3 | 34.0 | 363 | 73.8 |
| | ✓ | **88.4** ↑ 0.5 | **56.7** ↑ 1.7 | **42.0** ↑ 12.7 | **37.1** ↑ 3.1 | **381** ↑ 18 | **75.2** ↑ 1.4 |
| *Visual Encoder: SigLIP-ViT-SO400M/14@384* | | | | | | | |
| Vicuna-7B-v1.5 | – | 86.0 | 50.4 | 27.3 | 36.2 | 354 | 64.5 |
| | ✓ | **86.8** ↑ 0.8 | **53.2** ↑ 2.8 | **38.0** ↑ 10.7 | **41.6** ↑ 5.4 | **365** ↑ 11 | **65.7** ↑ 1.2 |
| Qwen2-7B-Instruct | – | 88.5 | 57.3 | 40.7 | 44.4 | 432 | 76.3 |
| | ✓ | **88.7** ↑ 0.2 | **58.2** ↑ 0.9 | **49.3** ↑ 8.6 | **46.3** ↑ 1.9 | **448** ↑ 16 | **76.9** ↑ 0.6 |

et al., 2023). Notably, even if $\mathbf{ROSS}^{\mathrm{R}}$-Latent (KL-16) has already outperformed the visual instruction tuning baseline by a large margin, $\mathbf{ROSS}^{\mathrm{D}}$ manages to bring significant improvements by replacing regression with denoising. Therefore, *denoising is better at handling visual spatial redundancy*.

Finally, we leverage the insights and conclusions from all our previous studies to train our $\mathbf{ROSS}$. Specifically, we regard the optimal formulation $\mathbf{ROSS}^{\mathrm{D}}$ (KL-16), *i.e.*, denoising with the KL-16 tokenizer, as our final $\mathbf{ROSS}$. Please refer to Appendix C.2 for ablations on the architecture of the denoiser, continuous tokenizer *v.s.* discrete tokenizer, and the denoising schedule.

## 5.2 IN-DEPTH ANALYSIS

**Attention Analysis.** We compute the attention scores of the *last* token with respect to *all visual tokens* on MMVP (Tong et al., 2024b). Quantitative and qualitative comparisons between the visual instruction tuning baseline (LLaVA) (Liu et al., 2024a) and our $\mathbf{ROSS}$ are provided in Table 1 and Figure 8, respectively. Table 1 reveals that the attention scores achieved by $\mathbf{ROSS}$ are *significantly higher* than those of LLaVA, indicating that the inclusion of vision-centric reconstructive objective $\mathcal{L}_{\mathrm{LMM}}^{\mathrm{visual}}$ effectively directs focus towards input images, thereby enhancing the comprehending visual signals. Similarly, Figure 8 demonstrate that the implementation of $\mathcal{L}_{\mathrm{LMM}}^{\mathrm{visual}}$ enables the alignment of attention closely with the *relevant visual elements* corresponding to the text query.

Table 1: **Quantitative comparison on attention values.** We conduct a T-test (Student, 1908) to compare the *means* and a Mann-Whitney U test (Mann & Whitney, 1947) to compare the *medians* of the two distributions. The mean and median of $\mathbf{ROSS}$ are both *significantly higher* than those of LLaVA.

| Statistic ($\times 10^{-4}$) | LLaVA | $\mathbf{ROSS}$ | P-value |
|------|-------|------|---------|
| Mean | 2.03 | 2.36 | $1.27 \times 10^{-7}$ |
| 25th Percentile | 1.50 | 1.81 | – |
| Median | 1.90 | 2.26 | $4.39 \times 10^{-9}$ |
| 75th Percentile | 2.42 | 2.76 | – |
| 95th Percentile | 3.51 | 3.69 | – |

**Generative *v.s.* Reconstructive.** We ablate the effectiveness of reconstruction over generation in Table 2. Similar to Sun et al. (2024b) and Dong et al. (2024), for generative cases, we adopt 576 learnable latent tokens to *query* the LMM and utilize the corresponding outputs as conditions to the denoiser. The detailed pipeline of these two methods can be found at Figure 11 in Appendix C.2. However, generative methods require *specific creation data* and can *not* be naively implemented on

Table 4: **Comparison to state-of-the-art LMMs**. A mixture of 2M caption data and 1.2M instruction tuning data are utilized for pre-training and fine-tuning, respectively. Our model outperforms them in most of the settings. We evaluate these models on: POPE (Li et al., 2023c) averaged accuracy, Hallu.: HallusionBench (Guan et al., 2024) average accuracy, MMB$^{EN}$: MMBench (Liu et al., 2023b) English dev split, MMB$^{CN}$: MMBench (Liu et al., 2023b) Chinese dev split, SEED$^I$: SEED-Bench-1 (Li et al., 2023a) with image accuracy, MMMU (Yue et al., 2024) validation split, MMVP (Tong et al., 2024b), GQA (Hudson & Manning, 2019) test-dev-balanced split, and AI2D (Hiippala et al., 2021) test split. $^\ddagger$We evaluate the official checkpoint/api using VLMEvalKit (Duan et al., 2024).

| Model | POPE | Hallu. | MMB$^{EN}$ | MMB$^{CN}$ | SEED$^I$ | MMMU | MMVP | GQA | AI2D |
|---|---|---|---|---|---|---|---|---|---|
| GPT-4V-1106 (OpenAI, 2023a) | 75.4 | 65.8$^\ddagger$ | 75.8 | 75.1$^\ddagger$ | 71.6 | 53.8 | 50.0 | 36.8 | 78.2 |
| Gemini-1.5 Pro (Team et al., 2023) | – | – | 73.6 | – | 70.7 | 47.9 | – | – | – |
| MM-1-8B (McKinzie et al., 2024) | 86.6 | – | 72.3 | – | 69.9 | 37.0 | – | 72.6 | – |
| Mini-Gemini-8B (Li et al., 2024f) | – | – | 72.7 | – | 73.2 | 37.3 | 18.7 | 64.5 | 73.5 |
| DeepSeek-VL-7B (Lu et al., 2024) | 85.8$^\ddagger$ | 44.1$^\ddagger$ | 73.2 | 72.8 | 70.4 | 36.6 | – | – | 64.9$^\ddagger$ |
| Cambrian-1-8B (Tong et al., 2024a) | 87.4$^\ddagger$ | 48.7$^\ddagger$ | 75.9 | 68.9$^\ddagger$ | **74.7** | 42.7 | 51.3 | 64.6 | 73.0 |
| **ROSS-7B** | **88.3** | **57.1** | **79.1** | **77.1** | 73.6 | **46.6** | **56.7** | **65.5** | **79.3** |
| *Base LLM: Vicuna-7B-v1.5* | | | | | | | | | |
| LLaVA-v1.5-7B$^\ddagger$ (Liu et al., 2024a) | 86.2 | 47.5 | 65.5 | 58.5 | 66.0 | 34.4 | 20.0 | 62.0 | 55.4 |
| LLaVA-v1.6-7B$^\ddagger$ (Liu et al., 2024b) | 86.5 | 35.8 | 67.4 | 60.1 | **70.2** | 35.8 | 37.3 | **64.2** | 67.1 |
| **ROSS-7B**$_{vicuna}$ | **88.2** | **55.2** | **67.7** | **61.3** | 67.6 | **36.9** | **39.3** | 63.7 | **69.3** |
| *Base LLM: Vicuna-13B-v1.5* | | | | | | | | | |
| LLaVA-v1.5-13B$^\ddagger$ (Liu et al., 2024a) | 82.5 | 44.9 | 68.8 | 63.6 | 68.2 | 36.6 | 32.0 | 63.3 | 60.8 |
| LLaVA-v1.6-13B$^\ddagger$ (Liu et al., 2024b) | 86.2 | 36.7 | 70.0 | 64.1 | 71.9 | 36.2 | 35.3 | **65.4** | 72.4 |
| Mini-Gemini-13B (Li et al., 2024f) | – | – | 68.6 | – | 73.2 | 37.3 | 19.3 | 63.7 | 70.1 |
| Cambrian-1-13B (Tong et al., 2024a) | 85.7$^\ddagger$ | 54.0$^\ddagger$ | **75.7** | 65.9$^\ddagger$ | **74.4** | 40.0 | 41.3 | 64.3 | 73.6 |
| **ROSS-13B**$_{vicuna}$ | **88.7** | **56.4** | 73.6 | **67.4** | 71.1 | **41.3** | **44.7** | 65.2 | **73.8** |

the original SFT data. To build creation data, we utilize GPT-4o to transfer 102K *caption* into *text-to-image* creation data from ShareGPT4V (Chen et al., 2023b) and combine them with the original SFT data. From Table 2, we can tell that reconstructive objectives boost comprehension while generative alternatives *cannot*. An intuitive explanation of this evidence can be found in Appendix C.2.

**ROSS with Different LLMs and Visual Encoders.** To demonstrate the effectiveness of our vision-centric supervision $\mathcal{L}_{LMM}^{visual}$ adopted by our **ROSS**, we conduct systematic experiments across different base LLMs and visual encoders. From Table 3, **ROSS** contributes to significant improvements *consistently*, especially on fine-grained comprehension benchmarks, *i.e.*, MMVP (Tong et al., 2024b) and ChartQA (Masry et al., 2022). Extended experiments on more representative benchmarks can be found at Table 12 in Appendix C.2.

**Reconstruction Results.** We fine-tune the denoiser to recover latent tokens from a frozen KL-16 provided by Rombach et al. (2022) conditioned on **ROSS-7B** features on ImageNet-1K (Deng et al., 2009) for *only five epochs*, where the denoiser manages to produce reasonable reconstruction results as illustrated at Figure 9 in Appendix C.1. This interesting finding demonstrates that high-level **ROSS-7B** features *actually contain image details*. We hope this finding will inspire future work.

**Computational Overhead.** The denoising process introduces a negligible increase in training time ($\approx$10% compared to the baseline), while the benefits outweigh the minor additional costs. Please refer to Table 10 in Appendix B for details.

## 5.3 COMPARISON WITH STATE-OF-THE-ARTS

**ROSS** utlizes a *single* SigLIP-ViT-SO400M/14@384 (Zhai et al., 2023) as the visual encoder. **ROSS-7B** utilizes Qwen2-7B-Instruct (Yang et al., 2024a) and **ROSS-13B**$_{vicuna}$ adopts Vicuna-13B-v1.5 (Chiang et al., 2023) as the base LLM. The implementation almost follows LLaVA-v1.5 (Liu et al., 2024a) *without* the high-resolution image-slicing technique (Liu et al., 2024b). Thus, our primary comparison of **ROSS** with alternative methods focuses on benchmarks that do *not* require exceptionally high-resolution inputs. We use a mixture of 2M caption data for the pre-training stage, which consists of 1246K from ShareGPT4V (Chen et al., 2023b) and 707K from ALLaVA (Chen et al., 2024a). The instruction tuning data is a mixture of Cambrian-737K (Tong et al., 2024a) and SMR-473K (Zhang

Table 5: **Transfer learning** on SpatialBench (Cai et al., 2024). "RGB" indicates using only RGB images for testing, while "RGB + D" represents taking depth maps as extra inputs. The performance of GPT-4o is obtained from Cai et al. (2024). *LMMs can better comprehend depth maps with $\mathcal{L}_{LMM}^{visual}$.*

| Method | Test Inputs | $\mathcal{L}_{LMM}^{visual}$ | MiDaS | Size | Reaching | Position | Existence | Counting | Average |
|---|---|---|---|---|---|---|---|---|---|
| LLaVA | RGB | – | – | 20.0 | 51.7 | 58.8 | 70.0 | 74.6 | 55.0 |
| | RGB + D | – | – | 21.7 ↑1.7 | 45.0 ↓6.7 | 58.8 – 0.0 | 65.0 ↓5.0 | 77.7 ↑3.1 | 53.6 ↓1.4 |
| | RGB | – | ✓ | 21.7 | 60.0 | 64.7 | 80.0 | 84.1 | 62.1 |
| | RGB + D | – | ✓ | 21.7 – 0.0 | 51.7 ↓8.3 | **70.6 ↑5.9** | 65.0 ↓15.0 | **91.1 ↑7.0** | 60.0 ↓2.1 |
| **ROSS** | RGB | ✓ | – | 25.0 | 53.3 | 64.7 | 70.0 | 75.3 | 57.7 |
| | RGB + D | ✓ | – | **28.3 ↑3.3** | **65.0 ↑11.7** | 67.6 ↑2.9 | **85.0 ↑15.0** | 84.6 ↑8.7 | **66.1 ↑8.4** |
| GPT-4o | RGB | – | – | 43.3 | 51.7 | 70.6 | 85.0 | 84.5 | 67.0 |
| | RGB + D | – | – | 40.0 ↓3.3 | 51.7 – 0.0 | 61.8 ↓8.8 | 90.0 ↑5.0 | 85.2 ↑0.7 | 65.7 ↓1.3 |

et al., 2024b). We further incorporate our **ROSS** with the "anyres" technique (Liu et al., 2024b) and compare with others on high-resolution benchmarks at Table 13 in Appendix C.3.

Illustrated in Table 4, we compare our **ROSS** with both private models (OpenAI, 2023a; Team et al., 2023; McKinzie et al., 2024) and open-sourced alternatives (Liu et al., 2024a;b; Tong et al., 2024a; Li et al., 2024f; Lu et al., 2024). The previous open-source state-of-the-art Cambrian-1 (Tong et al., 2024a) leverages *extrinsic assistance* that aggregates CLIP (Radford et al., 2021), SigLIP (Zhai et al., 2023), DINOv2 (Oquab et al., 2023), and ConvNext (Liu et al., 2022). On the other hand, our **ROSS** stands for *intrinsic activation*. With only a single SigLIP (Zhai et al., 2023) model as the visual encoder, our **ROSS** surpasses Cambrian-1 (Tong et al., 2024a), under most cases, *without* a careful choice of the visual experts and naturally maintains a lightweight inference procedure. **ROSS** is also data-efficient compared with Cambrian-1 (Tong et al., 2024a), since it requires 7M instruction tuning data. Notably, **ROSS-7B** even surpasses GPT-4V-1106 and Gemini-1.5 Pro on several benchmarks such as POPE (Li et al., 2023c), MMBench (Liu et al., 2023b), and MMVP (Tong et al., 2024b).

### 5.4 APPLICATIONS

**Transfer Learning on Understanding Depth Maps.** We further evaluate the transfer learning capability of our **ROSS** on SpatialBench (Cai et al., 2024), which requires the model to understand *depth maps*. We compare our **ROSS** with the visual instruction tuning baseline, with the *same* training data and model architecture. Also, we compare the effectiveness of the *extrinsic assistance* solution, *i.e.*, combining a depth expert MiDaS-3.0 (Birkl et al., 2023) to visual instruction tuning, with our *intrinsic activation* solution. Specifically, the pre-training data is LLaVA-558K (Liu et al., 2023a) and the fine-tuning data is SpatialQA-853K (Cai et al., 2024), where each conversation contains the RGB image and the *depth* maps extracted by ZoeDepth (Bhat et al., 2023). The visual encoder is CLIP-ViT-L/14@336 (Radford et al., 2021) and the base LLM is Qwen2-7B-Instruct (Yang et al., 2024a). As demonstrated in Table 5, our **ROSS** manages to make use of the extra depth map, as consistent and significant improvements are observed when taking "RGB + D" inputs for testing. Extrinsic assistance approaches *cannot* take advantage of extra depth maps when testing. Even GPT-4o *cannot* fully understand depth maps. Qualitative results can be found at Figure 16 in Appendix C.5.

## 6 CONCLUSION

This paper introduces reconstructive visual instruction tuning (**ROSS**), leveraging a vision-centric *reconstructive* objective to supervise visual outputs. To avoid being overwhelmed by heavily redundant raw RGB values, we train a denoiser to recover clean latent visual representations conditioning on visual outputs. Experimentally, the proposed objective indeed brings enhanced comprehension capabilities and reduced hallucinations. **ROSS** outperforms the state-of-the-art under most cases with only a *single* SigLIP (Zhai et al., 2023) as the visual encoder. The in-depth analysis demonstrates that high-level features from **ROSS-7B** actually contain sufficient details for low-level image reconstruction. This finding reveals the possibility of equipping comprehension LMMs with the ability of *naive generation* without the help of generation experts such as Stable Diffusion (Rombach et al., 2022).

## ACKNOWLEDGEMENTS

The work was supported by the National Science and Technology Major Project of China (No. 2023ZD0121300), the National Natural Science Foundation of China (No. U21B2042, No. 62320106010), the 2035 Innovation Program of CAS, and the InnoHK program.

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

APPENDIX

## A LATENT DIFFUSION MODELS

Given a set of clean latent tokens $z_0$ drawn from $p(z)$, the forward diffusion process is a Markov chain that gradually adds random Gaussian noise to the original sample:

$$q(z_t|z_{t-1}) = \mathcal{N}(\sqrt{1 - \beta_t}z_{t-1}, \beta_t\mathbf{I}), \tag{5}$$

where $\mathcal{N}(\boldsymbol{\mu}, \boldsymbol{\Sigma})$ denotes the Gaussian distribution, and $t$ indicates discrete timesteps. $\beta_t \in (0, 1)$ indicates a pre-defined time-dependent variance schedule. According to Ho et al. (2020), to admit sampling $z_t$ at an arbitrary timestep $t$ directly from $z_0$, this transition can be reformulated as

$$q(z_t|z_0) = \mathcal{N}(\sqrt{\bar{\alpha}_t}z_0, (1 - \bar{\alpha}_t)\mathbf{I}),$$
$$z_t = \sqrt{\bar{\alpha}_t}z_0 + \sqrt{1 - \bar{\alpha}_t}\boldsymbol{\epsilon}, \quad \boldsymbol{\epsilon} \sim \mathcal{N}(\mathbf{0}, \mathbf{I}), \tag{6}$$

where $\alpha_t = 1 - \beta_t$ and $\bar{\alpha}_t = \prod_{i=1}^{t} \alpha_t$. A latent diffusion model learns to *reverse* this progressive noise addition process for latent tokens. Specifically, to iteratively generate clean tokens $z_0$ from pure noise $z_T$ conditioned on $\mathcal{C}$, we need to reverse the forward process by

$$z_{t-1} = \frac{1}{\sqrt{\alpha_t}}\left(z_t - \frac{1 - \alpha_t}{\sqrt{1 - \bar{\alpha}_t}}\boldsymbol{\epsilon}_\pi(z_t; \mathcal{C}, t)\right) + \sigma_t\boldsymbol{\epsilon}, \tag{7}$$

where a $\pi$-parameterized neural network $\boldsymbol{\epsilon}_\pi$ is trained to predict the added noise during the forward process. $\sigma_t$ indicates the posterior noise variance. The training objective of $\boldsymbol{\epsilon}_\pi$ is

$$\mathcal{L}(\pi, z_0) = \mathbb{E}_{t, \boldsymbol{\epsilon}}\left[||\boldsymbol{\epsilon}_\pi(\sqrt{\bar{\alpha}_t}z_0 + \sqrt{1 - \bar{\alpha}_t}\boldsymbol{\epsilon}; \mathcal{C}, t) - \boldsymbol{\epsilon}||^2\right]. \tag{8}$$

## B IMPLEMENTATION DETAILS

Table 6: **Hyperparameters of ROSS.** We obtain most of the configurations from Liu et al. (2024a).

| Config | Stage I | Stage II |
|---|---|---|
| Trainable parts | projector + denoiser | projector + LLM + denoiser |
| Frozen parts | visual encoder + LLM + teacher tokenizer | visual encoder + teacher tokenizer |
| Global batch size | 256 | 128 |
| Batch size per GPU | 16 | 4 |
| Accumulated steps | 2 | 4 |
| DeepSpeed zero stage | 2 | 3 |
| Learning rate | $1\times10^{-3}$ | $2\times10^{-5}$ |
| Learning rate schedule | warmup + cosine decay | |
| Warmup ratio | 0.03 | |
| Weight decay | 0 | |
| Epoch | 1 | |
| Optimizer | AdamW | |
| Precision | bf16 | |

Table 7: **Details of the instruction tuning dataset** provided by Tong et al. (2024a).

| Dataset | # Samples |
|---|---|
| LLaVA (Liu et al., 2023a) | 158K |
| ShareGPT (Team, 2023) | 40K |
| VQAv2 (Goyal et al., 2017) | 83K |
| GQA (Hudson & Manning, 2019) | 72.1K |
| OKVQA (Marino et al., 2019) | 9K |
| OCRVQA (Mishra et al., 2019) | 80K |
| A-OKVQA (Schwenk et al., 2022) | 50K |
| TextVQA (Singh et al., 2019) | 21.9K |
| RefCOCO (Kazemzadeh et al., 2014) | 30K |
| VG (Krishna et al., 2017) | 86.4K |
| DVQA (Kafle et al., 2018) | 13K |
| DocVQA (Mathew et al., 2021) | 15K |
| ChartQA (Masry et al., 2022) | 28.1K |
| AI2 Diagrams (Kembhavi et al., 2016) | 15.5K |

Table 8: **Details of the instruction tuning dataset** provided by Zhang et al. (2024b).

| Dataset | # Samples |
|---|---|
| ScienccQA (Saikh et al., 2022) | 9K |
| TextbookQA (Kembhavi et al., 2017) | 9.5K |
| AI2 Diagrams (Kembhavi et al., 2016) | 12.4K |
| ChartQA (Masry et al., 2022) | 28.3K |
| DVQA (Kafle et al., 2018) | 200K |
| ArxivQA (Li et al., 2024d) | 100K |
| GeoQA3 (Chen et al., 2021) | 5K |
| Geometry3K (Lu et al., 2021) | 2.1K |
| GeoQA+ (Cao & Xiao, 2022) | 72.3K |
| MathVision (Wang et al., 2024d) | 2.7K |
| TabMWP (Lu et al., 2022) | 30.7K |

Table 9: **Summary of the evaluation benchmarks.** Prompts are mostly borrowed from VLMEvalKit (Duan et al., 2024) and lmms-eval (Li et al., 2024b).

| Benchmark | Response formatting prompts |
|---|---|
| POPE (Li et al., 2023c) | – |
| HallusionBench (Guan et al., 2024) | Answer the question using a single word or phrase. |
| MMBench (Liu et al., 2023b) | Answer with the option's letter from the given choices directly. |
| SEED-Bench (Li et al., 2023a) | Answer with the option's letter from the given choices directly. |
| MMMU (Yue et al., 2024) | Answer with the option's letter from the given choices directly. |
| MMVP (Tong et al., 2024b) | Answer with the option's letter from the given choices directly. |
| AI2D (Hiippala et al., 2021) | Answer with the option's letter from the given choices directly. |
| RealWorldQA (xAI, 2024) | Answer with the option's letter from the given choices directly. |
| GQA (Hudson & Manning, 2019) | Answer the question using a single word or phrase. |
| ChartQA (Masry et al., 2022) | Answer the question using a single word or phrase. |
| OCRBench (Liu et al., 2023c) | Answer the question using a single word or phrase. |
| DocVQA (Mathew et al., 2021) | Answer the question using a single word or phrase. |
| InfoVQA (Biten et al., 2022) | Answer the question using a single word or phrase. |
| TextVQA (Singh et al., 2019) | Answer the question using a single word or phrase. |

Table 10: **Comparisons on computational costs during the instruction tuning stage** with Cambrian-737K (Tong et al., 2024a), where evaluations are conducted using 8 A100 GPUs with a global batch size of 128. Due to the limited GPU memory, we accumulate 4 gradient steps and the batch size per GPU is 4. The whole stage requires 5757 training steps. GPU memories are averaged over 8 GPUs with DeepSpeed Zero 3.

| Vision | Base LLM | $\mathcal{L}_{\text{LMM}}^{\text{visual}}$ | Trainable Parameters | Speed (s/iter) | Time | GPU Memory |
|---|---|---|---|---|---|---|
| CLIP-L/336 | Qwen2-7B-Instruct | – | 7.63 B | 8.31 | 13h 17min | 45.34 G |
| CLIP-L/336 | Qwen2-7B-Instruct | ✓ | 7.68 B | 9.02 (1.09×) | 14h 25min | 46.62 G (1.03×) |
| CLIP-L/336 | Vicuna-13B-v1.5 | – | 13.05 B | 13.33 | 21h 19min | 48.62 G |
| CLIP-L/336 | Vicuna-13B-v1.5 | ✓ | 13.11 B | 14.69 (1.10×) | 23h 30min | 49.07 G (1.01×) |
| SigLIP-L/384 | Qwen2-7B-Instruct | – | 7.63 B | 8.77 | 14h 1min | 47.08 G |
| SigLIP-L/384 | Qwen2-7B-Instruct | ✓ | 7.68 B | 9.48 (1.08×) | 15h 9min | 52.07 G (1.11×) |
| SigLIP-L/384 | Vicuna-13B-v1.5 | – | 13.05 B | 14.22 | 22h 44min | 48.80 G |
| SigLIP-L/384 | Vicuna-13B-v1.5 | ✓ | 13.11 B | 15.32 (1.08×) | 24h 30min | 52.68 G (1.08×) |

**Hyperparameters.** The hyperparameters of ROSS are provided in Table 6. We simply borrow most configurations from LLaVA-v1.5 (Liu et al., 2024a) without further tuning, as we find it works well with our ROSS, even if we adopt SigLIP (Zhai et al., 2023) and Qwen2 (Yang et al., 2024a) while the original LLaVA-v1.5 (Liu et al., 2024a) utilized CLIP (Radford et al., 2021) and Vicuna-v1.5 (Chiang et al., 2023). As SigLIP represents a single $384 \times 384$ image with 729 tokens and the downsampling ratio of the teacher tokenizer KL-16 (Rombach et al., 2022) is 16, we set the input resolution of the teacher tokenizer as $432 = \sqrt{729} \times 16$ to produce 729 fine-grained tokens as denoising targets.

**Instruction Tuning Data.** When comparing with state-of-art LMMs in Table 4, our ROSS is trained on approximately 1.2M instruction tuning data, which is a mixture of Cambrian-737K (Tong et al., 2024a) and SMR-473K (Zhang et al., 2024b). Details of these two instruction tuning datasets are listed in Table 7 and Table 8, respectively. There might be some overlap but we simply concat these two datasets as it is already empirically effective.

**Evaluation Prompts.** We provide a thorough examination of all evaluation benchmarks utilized in this paper in Table 9. Notably, for MMVP (Tong et al., 2024b), which is not officially supported by VLMEvalKit (Duan et al., 2024), we follow Cambrian-1 (Tong et al., 2024a) to reformat the original question into a multiple-choice format and compute the accuracy using exact matching.

**Computational Costs.** As demonstrated in Table 10, the denoising process introduces a negligible increase in training time ($\approx 10\%$ compared to the baseline), while the benefits outweigh the minor additional costs.

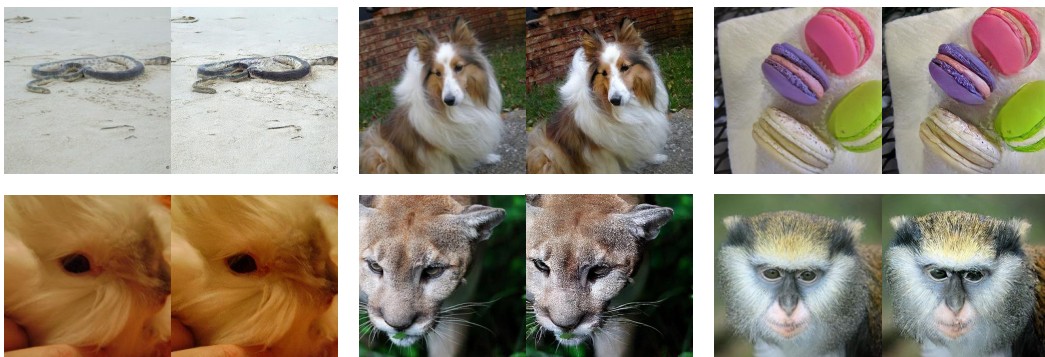

Figure 9: **Reconstruction results** on ImageNet-1K (Deng et al., 2009) validation set. For each tuple, we show the input image (left) and the reconstructed image (right). Reasonable reconstruction results demonstrate that *high-level features of* **ROSS-7B** *can be projected back into the pixel space.*

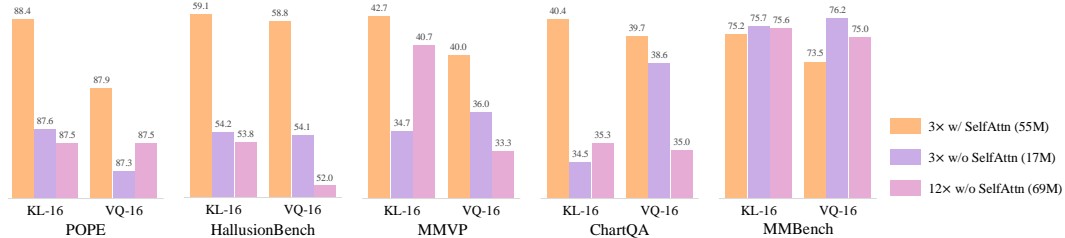

Figure 10: **The architecture of the denoiser and the choice of fine-grained tokenizer.** The self-attention module illustrated in Figure 4b is crucial since orange bars consistently outperform others on hallucination and fine-grained comprehension benchmarks, while maintaining similar performances on the general understanding benchmark. KL-16 provided by Rombach et al. (2022) is better than VQ-16 provided by Sun et al. (2024a), as quantization may lead to information loss.

## C MORE EXPERIMENTS

### C.1 RECONSTRUCTION RESULTS

We fine-tune the denoiser to recover latent tokens from a frozen KL-16 provided by Rombach et al. (2022) conditioned on frozen **ROSS-7B** features on ImageNet-1K (Deng et al., 2009) for *only five epochs*, where the denoiser manages to produce reasonable reconstruction results as illustrated in Figure 9. This interesting finding demonstrates that high-level **ROSS-7B** features *actually contain image details*.

### C.2 MORE ABLATIONS

**KL-16 *v.s.* VQ-16.** Our default tokenizer is a continuous VAE (Kingma, 2013) with Kullback-Leibler (KL) divergence trained by Rombach et al. (2022). We further conduct experiments with a *discrete* tokenizer provided by Sun et al. (2024a), which is a VQGAN (Esser et al., 2021), *i.e.*, VQVAE (Oord et al., 2017) with additional perceptual loss (Zhang et al., 2018) and adversarial loss (Goodfellow et al., 2014). As demonstrated in Figure 10, *KL-16 outperforms VQ-16*. One intuitive explanation is that KL-16 preserves more low-level details than VQ-16 since quantization may lead to information loss. Moreover, quantitatively, on ImageNet (Deng et al., 2009) 256×256 validation set, KL-16 achieves 0.87 rFID (Heusel et al., 2017) while the rFID (Heusel et al., 2017) of VQ-16 is 2.19.

**Architecture of the Denoiser.** Illustrated in Figure 10, *the self-attention module is crucial*, as original visual outputs $x_{i \leq N}$ are actually *causal* and we need to model inter-token discrepancy via self-attention. The number of trainable parameters is *not* the crux.

**Schedule of $\beta$.** We study the effectiveness of different schedules of $\beta$ in Table 11. From the table, we can tell that even with different schedules of $\beta$, **ROSS** *consistently* improves the baseline, demonstrating its robustness to the denoising schedule.

Table 11: **Ablations on different schedules of** $\beta$. ROSS *consistently* improves the baseline, demonstrating its robustness to the denoising schedule.

| Schedule of $\beta$ | POPE | Hallu. | MMVP | ChartQA | MMB$^{EN}$ |
|---|---|---|---|---|---|
| – | 87.9 | 55.0 | 29.3 | 34.0 | 73.8 |
| Linear (Ho et al., 2020) | 88.1 ↑ 0.2 | 57.3 ↑ 2.3 | 42.0 ↑ 12.4 | 39.2 ↑ 5.2 | 75.1 ↑ 1.3 |
| Scaled Linear (Rombach et al., 2022) | **88.4** ↑ **0.5** | 58.3 ↑ 3.3 | 40.0 ↑ 10.4 | **40.7** ↑ **6.7** | 75.3 ↑ 1.5 |
| GLIDE Softmax (Nichol et al., 2022) | **88.4** ↑ **0.5** | **59.1** ↑ **4.1** | **42.7** ↑ **13.4** | 40.4 ↑ 6.4 | 75.2 ↑ 1.4 |
| GeoDiff Sigmoid (Xu et al., 2022) | 88.2 ↑ 0.3 | 57.7 ↑ 2.7 | 41.3 ↑ 11.7 | 38.9 ↑ 4.9 | **75.5** ↑ **1.7** |

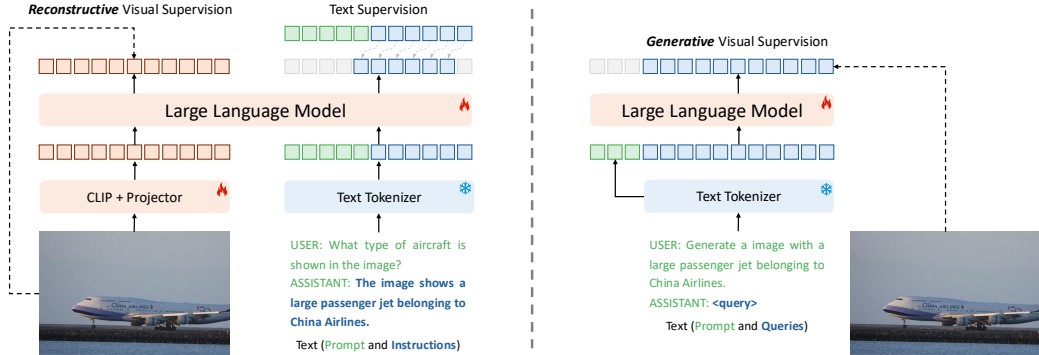

Figure 11: **Pipeline comparison between reconstructive and generative.** The reconstructive objective (left) does *not* require specific data formulations and can be easily combined with current visual instruction tuning data. However, the generative objective (right) needs specific *text-to-image* creation data, which could be converted by image-to-text caption data.

**Generative *v.s.* Reconstructive.** We offer a detailed pipeline comparison in Figure 11. Experimental results have already been provided in Table 2. The implementation of generative methods is similar to Sun et al. (2024b) and Dong et al. (2024), where we adopt 576 learnable queries as inputs and take the corresponding outputs as conditions for the denoiser.

We hypothesize that the underlying reason for the lower performance of generative methods in comprehension tasks is *the weak correspondence between inputs and supervision* under generative settings, which typically arises from both the (1) data and the (2) design of these methods.

(1) Typical generative methods that explore the synergy of comprehension and generation, usually leverage image generation conditioned on text instructions on *(i) text-to-image datasets* or *(ii) interleaved datasets* as extra supervision. However, (i) text-to-image datasets are typically designed to generate *high-aesthetic* samples rather than text-aligned ones, and (ii) interleaved datasets aim to enable few-shot learning via interleaving independent supervised examples, where reasoning becomes more important than alignment. Therefore, there exists a clear disconnect where the supervision (image) has little to do with the input (text instruction). For example, the CLIP-Score (Hessel et al., 2021), which measures the similarity between text and images, is only 0.3043 for the LAION-Art dataset (Schuhmann et al., 2022) and 0.2842 for the MMC4 dataset (Zhu et al., 2023), indicating that these datasets are *not* well-suited for tasks requiring strong text-image alignment.

(2) Even when we attempt to ensure image-text alignment by converting aligned caption data into creation data for supervision, the results demonstrated in Table 2 remain unsatisfactory. This suggests that the *design of generative objectives itself does not inherently require a strong correspondence* between inputs and supervision targets.

In contrast, reconstructive methods like **ROSS** leverage the original input images as auxiliary supervision, ensuring a strong and direct correspondence, which is crucial for tasks requiring accurate comprehension and interpretation of multimodal data, leading to significantly improved performance.

**Extended Ablations on Different LLMs and Visual Encoders.** We extend the ablation in Table 3 by incorporating more benchmarks, providing a more balanced and representative distribution of tasks. Empirical results in Table 12 demonstrate that our proposed vision-centric supervision utilized by **ROSS** leads to significant improvements in most cases. Moreover, we found **ROSS** contributes more

Table 12: **Extended ablations** on The effectiveness of the vision-centric supervision $\mathcal{L}_{\text{LMM}}^{\text{visual}}$ among various LLMs and visual encoders. Pre-training data is LLaVA-558K (Liu et al., 2023a) and instruction tuning data is Cambrian-737K (Tong et al., 2024a). Evaluations of POPE (Li et al., 2023c), HallusionBench (Guan et al., 2024), MMBench (Liu et al., 2023b), SEED-Bench-1 (Li et al., 2023a), MMMU (Yue et al., 2024), MMVP (Tong et al., 2024b), AI2D (Hiippala et al., 2021), OCRBench (Liu et al., 2023c), and RealWorldQA (xAI, 2024) are conducted with VLMEvalKit (Duan et al., 2024), while evaluations of ChartQA (Masry et al., 2022), DocVQA (Mathew et al., 2021), InfoVQA (Biten et al., 2022), and TextVQA (Singh et al., 2019) are conducted with lmms-eval (Li et al., 2024b).

| | CLIP-ViT-L/14@336 | | | | SigLIP-ViT-SO400M/14@384 | | | |
| | Vicuna-7B-v1.5 | | Qwen2-7B-Instruct | | Vicuna-7B-v1.5 | | Qwen2-7B-Instruct | |
| Benchmark | LLaVA | ROSS | LLaVA | ROSS | LLaVA | ROSS | LLaVA | ROSS |
|---|---|---|---|---|---|---|---|---|
| POPE$_{\text{acc}}$ | 86.3 | **87.2** ↑ 0.9 | 87.9 | **88.4** ↑ 0.5 | 86.0 | **87.7** ↑ 1.7 | 88.5 | **88.7** ↑ 0.2 |
| HallusionBench$_{\text{aAcc}}$ | 52.5 | **55.8** ↑ 3.3 | 55.0 | **59.1** ↑ 4.1 | 50.4 | **53.8** ↑ 3.4 | 57.3 | **58.2** ↑ 0.9 |
| MMBench-EN$_{\text{dev}}$ | 67.0 | **67.6** ↑ 0.6 | 73.8 | **75.2** ↑ 1.4 | 64.5 | **69.2** ↑ 4.7 | 76.3 | **76.9** ↑ 0.6 |
| MMBench-CN$_{\text{dev}}$ | **60.0** | 59.8 ↓ 0.2 | 72.9 | **73.7** ↑ 0.8 | 63.1 | **63.4** ↑ 0.3 | 75.7 | **76.3** ↑ 0.7 |
| SEED$_{\text{img}}$ | **66.7** | 66.4 ↓ 0.3 | 70.3 | **70.7** ↑ 0.4 | 68.2 | **69.0** ↑ 0.8 | **72.3** | 72.1 ↓ 0.2 |
| MMMU$_{\text{dev}}$ | 30.0 | **34.0** ↑ 4.0 | 44.0 | **45.3** ↑ 1.3 | 33.3 | **38.0** ↑ 4.7 | 38.7 | **41.3** ↑ 2.6 |
| MMMU$_{\text{val}}$ | 35.3 | **36.0** ↑ 0.7 | 41.9 | **42.6** ↑ 0.7 | 34.2 | **35.4** ↑ 1.2 | 41.8 | **43.8** ↑ 2.0 |
| MMVP | 28.0 | **36.0** ↑ 8.0 | 29.3 | **42.7** ↑ 13.4 | 27.3 | **38.0** ↑ 10.7 | 40.7 | **49.3** ↑ 8.6 |
| AI2D$_{\text{test}}$ | 61.2 | **61.4** ↑ 0.2 | 71.9 | **73.3** ↑ 1.4 | **62.6** | 62.4 ↓ 0.2 | 74.0 | **74.5** ↑ 0.5 |
| ChartQA$_{\text{test}}$ | 32.9 | **39.8** ↑ 6.9 | 36.2 | **41.6** ↑ 5.4 | 34.0 | **48.2** ↑ 14.2 | 44.4 | **46.9** ↑ 2.5 |
| DocVQA$_{\text{val}}$ | 33.4 | **41.6** ↑ 8.2 | 31.1 | **44.7** ↑ 13.6 | 40.4 | **40.7** ↑ 0.3 | 39.2 | **39.3** ↑ 0.1 |
| InfoVQA$_{\text{val}}$ | 21.2 | **26.4** ↑ 5.2 | 22.1 | **39.3** ↑ 16.2 | 22.8 | **23.3** ↑ 0.5 | 24.0 | **25.1** ↑ 1.1 |
| TextVQA$_{\text{val}}$ | 55.7 | **58.7** ↑ 3.0 | 52.0 | **54.1** ↑ 2.1 | 60.5 | **62.6** ↑ 2.1 | 56.3 | **57.5** ↑ 1.2 |
| OCRBench | 339 | **350** ↑ 11 | 363 | **381** ↑ 18 | 354 | **365** ↑ 11 | 432 | **448** ↑ 16 |
| RealWorldQA | 52.7 | **53.2** ↑ 0.5 | 56.7 | **57.4** ↑ 0.7 | 55.0 | **57.1** ↑ 2.1 | 57.9 | **59.1** ↑ 1.2 |
| Average | 47.8 | **50.6** ↑ 2.8 | 52.1 | **56.4** ↑ 4.3 | 49.2 | **52.4** ↑ 3.2 | 55.4 | **56.9** ↑ 1.5 |

Table 13: **Comparison to state-of-the-art LMMs on benchmarks requires high-resolution inputs.** We evaluate models on: ChartQA (Masry et al., 2022), DocVQA (Mathew et al., 2021) val set, InfoVQA (Biten et al., 2022) val set, TextVQA (Singh et al., 2019) val set, OCRBench (Liu et al., 2023c), and RealWorldQA (xAI, 2024). [‡]We evaluate the official checkpoint.

| Model | ChartQA | DocVQA | InfoVQA | TextVQA | OCRBench | RealWorldQA |
|---|---|---|---|---|---|---|
| GPT-4V-1106 (OpenAI, 2023a) | 78.5 | 88.4 | – | 78.0 | 645 | 61.4 |
| Gemini-1.5 Pro (Team et al., 2023) | 81.3 | 86.5 | – | 78.1 | – | 67.5 |
| Grok-1.5 (xAI, 2024) | 76.1 | 85.6 | – | 78.1 | – | 68.7 |
| LLaVA-v1.5-7B[‡] (Liu et al., 2024a) | 18.2 | 28.1 | 25.7 | 58.2 | 317 | 54.9 |
| LLaVA-v1.6-7B[‡] (Liu et al., 2024b) | 65.5 | 74.4 | 37.1 | 64.8 | 532 | 57.6 |
| Cambrian-1-8B (Tong et al., 2024a) | 73.3 | 77.8 | – | 71.7 | **624** | 64.2 |
| **ROSS-7B**$_{\text{anyres}}$ | **76.9** | **81.8** | **50.5** | 72.2 | 607 | **66.2** |

significant improvements over fine-grained comprehension datasets, such as HallusionBench (Guan et al., 2024), MMVP (Tong et al., 2024b), and ChartQA (Masry et al., 2022).

## C.3 COMPARISON ON HIGH-RESOLUTION BENCHMARKS

We incorporate the "anyres" technique proposed by LLaVA-v1.6 (Liu et al., 2024b) into our **ROSS**. Specifically, for each image, we employ a grid configuration of $384 \times \{2 \times 2, 1 \times \{2,3,4\}, \{2,3,4\} \times 1\}$ to identify the input resolution, resulting in a maximum of $5 \times 729 = 3,645$ visual tokens. *Each $384 \times 384$ crop is required to reconstruct the original input via the denoising objective proposed by* **ROSS**. In Table 13, our **ROSS-7B**$_{\text{anyres}}$ surpasses LLaVA-v1.6-7B (Liu et al., 2024b) and Cambrian-1-8B (Tong et al., 2024a) under most cases. These results indicate that **ROSS** not only performs well at lower resolutions but also maintains its competitive edge at higher resolutions, making it a robust and versatile method.

Table 14: **Evaluations on language performance.** We evaluate multi-modal benchmarks that mainly require general knowledge following Tong et al. (2024a). Furthermore, we incorporate representative language benchmarks, including general understanding on MMLU (Hendrycks et al., 2020) and HellaSwag (Zellers et al., 2019), and instruction-following on IFEval (Zhou et al., 2023). ROSS does *not* harm language capabilities as it brings improvements in most cases.

| | CLIP-ViT-L/14@336 | | | | SigLIP-ViT-SO400M/14@384 | | | |
| | Vicuna-7B-v1.5 | | Qwen2-7B-Instruct | | Vicuna-7B-v1.5 | | Qwen2-7B-Instruct | |
| Benchmark | LLaVA | **ROSS** | LLaVA | **ROSS** | LLaVA | **ROSS** | LLaVA | **ROSS** |
|---|---|---|---|---|---|---|---|---|
| *Vision-Language Benchmarks on Knowledge* | | | | | | | | |
| ScienceQA$_{test}$ | 68.5 | **69.0** ↑ 0.5 | 76.5 | **77.4** ↑ 0.9 | 69.6 | **71.3** ↑ 1.7 | 78.3 | **78.5** ↑ 0.2 |
| MMMU$_{dev}$ | 30.0 | **34.0** ↑ 4.0 | 44.0 | **45.3** ↑ 1.3 | 33.3 | **38.0** ↑ 4.7 | 38.7 | **41.3** ↑ 2.6 |
| MMMU$_{val}$ | 35.3 | **36.0** ↑ 0.7 | 41.9 | **42.6** ↑ 0.7 | 34.2 | **35.4** ↑ 1.2 | 41.8 | **43.8** ↑ 2.0 |
| AI2D$_{test}$ | 61.2 | **61.4** ↑ 0.2 | 71.9 | **73.3** ↑ 1.4 | **62.6** | 62.4 ↓ 0.2 | 74.0 | **74.5** ↑ 0.5 |
| *Language Benchmarks* | | | | | | | | |
| MMLU | 26.5 | **27.4** ↑ 0.9 | 57.1 | **60.7** ↑ 3.6 | **26.0** | 25.9 ↓ 0.1 | 60.9 | **61.0** ↑ 0.1 |
| HellaSwag$_{acc-norm}$ | **27.0** | 26.9 ↓ 0.1 | 46.4 | 46.2 ↓ 0.2 | **27.1** | 27.0 ↓ 0.1 | 45.5 | **46.6** ↑ 1.1 |
| IFEval$_{strict-inst}$ | 41.2 | **44.6** ↑ 3.4 | 47.1 | **49.2** ↑ 2.1 | 43.6 | **43.8** ↑ 0.2 | 47.8 | **48.1** ↑ 0.3 |
| IFEval$_{strict-prompt}$ | 28.7 | **35.3** ↑ 6.7 | 35.1 | **37.0** ↑ 1.9 | 32.5 | **33.1** ↑ 0.6 | 35.3 | **36.2** ↑ 0.9 |
| Average | 39.8 | **41.8** ↑ 2.0 | 52.5 | **54.0** ↑ 1.5 | 41.1 | **42.1** ↑ 1.0 | 52.8 | **53.8** ↑ 1.0 |

Table 15: **Model scaling of ROSS.** We take Qwen2.5 series (Team, 2024) as the base language model and CLIP-ViT-L/14@336 (Radford et al., 2021) as the visual encoder. Pre-training data is LLaVA-558K (Liu et al., 2023a) and the instruction tuning data is LLaVA-665K (Liu et al., 2024a). **ROSS** *brings improvements over the baseline across different model sizes in most cases.*

| Benchmark | 0.5B | | 1.5B | | 3B | | 7B | |
| | LLaVA | **ROSS** | LLaVA | **ROSS** | LLaVA | **ROSS** | LLaVA | **ROSS** |
|---|---|---|---|---|---|---|---|---|
| POPE$_{acc}$ | 50.0 | **60.4** ↑ 10.4 | 85.3 | **87.9** ↑ 2.4 | 87.3 | **88.1** ↑ 0.8 | 87.9 | **88.4** ↑ 0.5 |
| HallusionBench$_{aAcc}$ | 45.8 | **48.0** ↑ 2.2 | 48.7 | **49.6** ↑ 0.9 | 52.2 | 52.2 – 0.0 | 48.7 | **53.7** ↑ 5.0 |
| MMBench-EN$_{dev}$ | 55.2 | **60.4** ↑ 5.2 | 67.5 | **68.2** ↑ 1.7 | 70.6 | **71.4** ↑ 0.8 | 75.0 | **75.7** ↑ 0.7 |
| MMBench-CN$_{dev}$ | 45.6 | **48.9** ↑ 3.3 | 62.4 | **63.9** ↑ 1.5 | 68.0 | **69.1** ↑ 1.1 | **73.6** | 73.5 ↓ 0.1 |
| SEED$_{img}$ | **55.8** | 55.6 ↓ 0.2 | 66.3 | **66.8** ↑ 0.5 | 68.2 | **68.4** ↑ 0.2 | 70.6 | **71.0** ↑ 0.4 |
| OCRBench | 229 | **248** ↑ 19 | 291 | **298** ↑ 7 | **313** | 308 ↓ 5 | 334 | **358** ↑ 24 |
| MMMU$_{dev}$ | 35.2 | **36.0** ↑ 0.8 | 44.7 | **45.0** ↑ 0.3 | 48.7 | **49.0** ↑ 0.3 | 48.0 | 48.0 – 0.0 |
| MMMU$_{val}$ | 38.0 | **40.3** ↑ 1.7 | 41.8 | **43.6** ↑ 1.8 | 41.6 | **42.7** ↑ 1.1 | 47.3 | **48.0** ↑ 0.7 |
| AI2D$_{test}$ | 45.3 | **46.0** ↑ 0.7 | 59.0 | **59.5** ↑ 0.5 | 62.9 | **63.2** ↑ 0.3 | 68.3 | **68.5** ↑ 0.2 |
| RealWorldQA | 45.1 | **46.4** ↑ 1.3 | 50.5 | **53.5** ↑ 3.0 | 55.7 | **57.9** ↑ 2.2 | 59.5 | **59.9** ↑ 0.4 |
| Average | 43.9 | **46.7** ↑ 2.8 | 55.3 | **56.8** ↑ 1.5 | 58.9 | **59.3** ↑ 0.4 | 61.2 | **62.3** ↑ 1.1 |

Table 16: **Data scaling of ROSS.** We take Qwen2-7B-Instruct (Yang et al., 2024a) as the base language model and CLIP-ViT-L/14@336 (Radford et al., 2021) as the visual encoder. **ROSS** *consistently brings significant improvements as the training data scale increases.*

| PT | SFT | $\mathcal{L}_{LMM}^{visual}$ | POPE | Hallu. | ChartQA | OCRBench | MMB$^{EN}$ | AI2D |
|---|---|---|---|---|---|---|---|---|
| 558K | 737K | − | 87.9 | 55.0 | 34.0 | 363 | 73.8 | 72.4 |
| | | ✓ | **88.4** ↑ 0.5 | **59.1** ↑ 4.1 | **40.4** ↑ 6.4 | **380** ↑ 17 | **75.2** ↑ 1.4 | **73.3** ↑ 0.9 |
| 558K | 1.2M | − | 88.5 | 57.3 | 37.0 | 389 | 75.7 | 74.5 |
| | | ✓ | **88.8** ↑ 0.3 | **57.8** ↑ 0.5 | **42.0** ↑ 5.0 | **392** ↑ 3 | **76.8** ↑ 1.1 | **74.7** ↑ 0.2 |
| 2M | 737K | − | 88.1 | 55.6 | 37.3 | 384 | 76.2 | 72.3 |
| | | ✓ | **88.3** ↑ 0.2 | **56.2** ↑ 0.6 | **41.9** ↑ 4.5 | **398** ↑ 14 | **77.0** ↑ 0.8 | **73.4** ↑ 1.1 |
| 2M | 1.2M | − | 88.5 | 53.8 | 41.2 | 388 | 76.5 | 73.9 |
| | | ✓ | **88.9** ↑ 0.4 | **57.3** ↑ 2.5 | **43.2** ↑ 2.0 | **405** ↑ 17 | **78.0** ↑ 1.5 | **74.1** ↑ 0.2 |

## C.4 MORE ANALYSIS

**Language Capabilities.** One possible concern of **ROSS** is that this type of low-level reconstruction may harm the high-level language capabilities. To investigate this issue, we evaluate multi-modal

From the camera's perspective, is the spider web very dense or relatively sparse?
(a) Very dense          **(b) Relatively sparse**

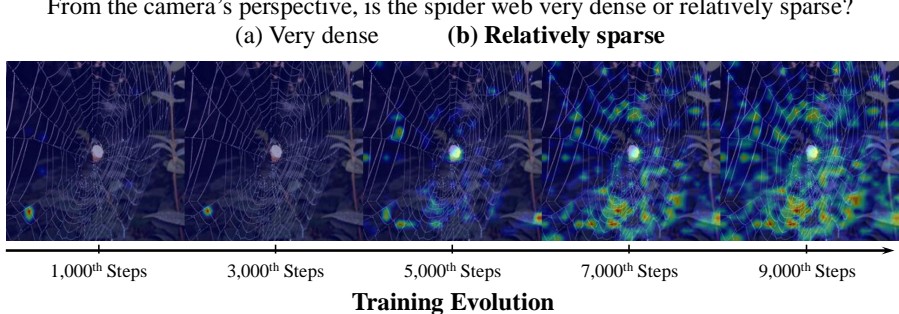

Figure 12: **Qualitative comparison using GradCAM (Selvaraju et al., 2020)** on MMVP (Tong et al., 2024b). We visualize the gradient of the second-to-last block of the LMM using the option of the ground-truth answer as the target class. Equipped with our proposed vision-centric supervision signals, **ROSS** provides more reasonable gradients and urges LMMs to focus on relevant regions (the spider web) as the training evolves.

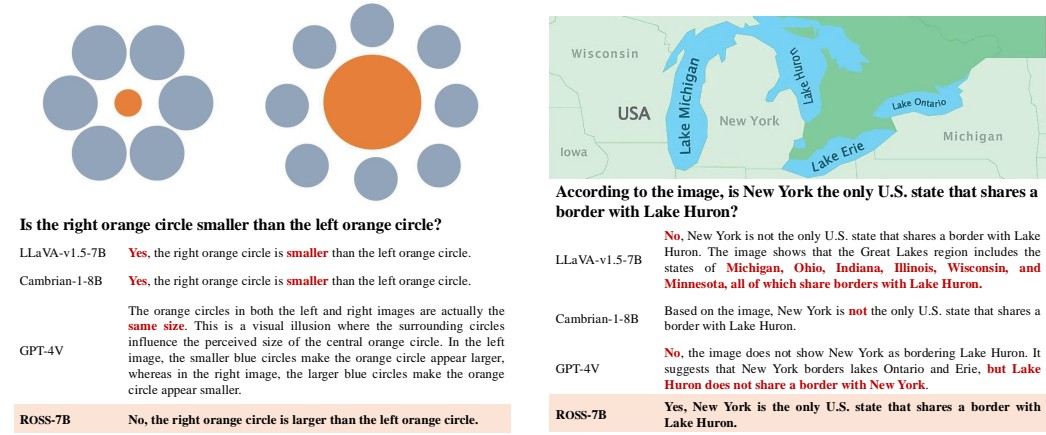

Figure 13: **Qualitative comparisons on HallusionBench (Guan et al., 2024).**

benchmarks that mainly require general knowledge following (Tong et al., 2024a), including ScienceQA (Saikh et al., 2022), MMMU (Yue et al., 2024), and AI2D (Hiippala et al., 2021). Furthermore, we incorporate representative language benchmarks, including general understanding on MMLU (Hendrycks et al., 2020) and HellaSwag (Zellers et al., 2019), and instruction-following on IFEval (Zhou et al., 2023). Empirical results in Table 14 demonstrate that **ROSS** does *not* harm language capabilities as it brings improvements in most cases.

**Model Scaling Properties.** To study the stability and scalability of **ROSS** across different model sizes, we use the Qwen2.5 series (Team, 2024) with varying sizes as the base language model while keeping the CLIP-ViT-L/14@336 (Radford et al., 2021) as the visual encoder. The pre-training data is LLaVA-558K (Liu et al., 2023a), and the instruction tuning data is LLaVA-665K (Liu et al., 2024a). The results, shown in Table 15, demonstrate that **ROSS** *brings improvements over the baseline (LLaVA) across different model sizes in most cases*.

**Data Scaling Properties.** To study the impact of the training data scale, we used Qwen2-7B-Instruct (Yang et al., 2024a) as the base language model and CLIP-ViT-L/14@336 (Radford et al., 2021) as the visual encoder. We compared the performance of **ROSS** and the baseline under different scales of training data. Table 16 demonstrates that **ROSS** *consistently brings significant improvements as the training data scale increases*.

**Gradient Analysis.** To better explain the reasoning behind how the vison-centric supervision enables the model to focus on relevant areas of the image during VQA tasks, we provide qualitative comparison using GradCAM (Selvaraju et al., 2020) on MMVP (Tong et al., 2024b) in Figure 12, since GradCAM helps in understanding which parts of the image the model is focusing on, making the model's decision-making process more transparent. In our analysis, we visualize the gradient of

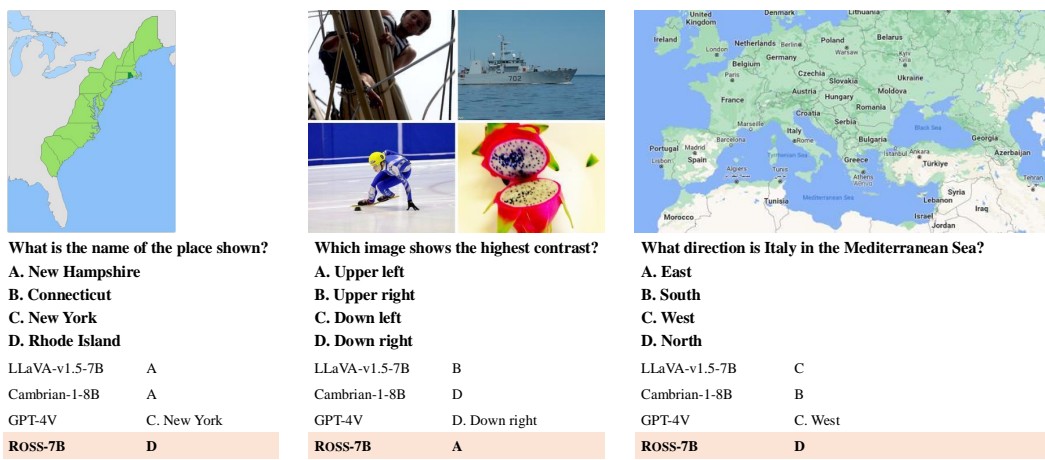

Figure 14: **Qualitative comparisons on MMbench (Guan et al., 2024) English dev split.**

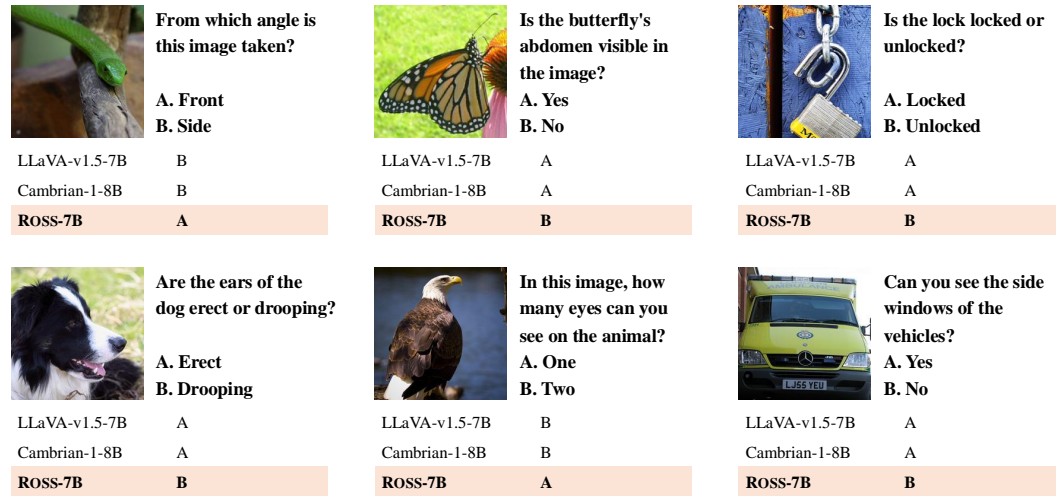

Figure 15: **Qualitative comparisons on MMVP (Tong et al., 2024b).**

the second-to-last block of the LMM, regarding the option of the ground-truth answer as the target class. Specifically in this case, where the providing question is about the spider web, our proposed vision-centric supervision signals provide more reasonable gradients and urge LMMs to focus on relevant regions, *i.e.*, the spider web, as the training evolves.

## C.5 QUALITATIVE COMPARISONS

We provide sufficient qualitative comparisons in Figure 13, Figure 14, Figure 15, and Figure 16 on HallusionBench (Guan et al., 2024), MMBench (Liu et al., 2023b) English dev split, MMVP (Tong et al., 2024b), and SpatialBench (Cai et al., 2024), respectively. In Figure 13, Figure 14, and Figure 15, we compare our **ROSS-7B** with the instruction tuning baseline LLaVA-v1.5-7B (Liu et al., 2024a), the state-of-the-art open-source method using extrinsic assistance Cambrian-1-8B (Tong et al., 2024a), and GPT-4V (OpenAI, 2023a).

As demonstrated in Figure 13, where we highlight the *wrong* parts of each prediction in red, our **ROSS** manages to correctly answer the question with reduced hallucinations even when GPT-4V fails. Cambrian-1 (Tong et al., 2024a) even fails to follow the instructions in the second example. This could be because a super huge SFT data (7M) may harm the instruction-following abilities of LMMs. Qualitative results shown in Figure 14 demonstrate both enhanced reasoning abilities (the first example), low-level comprehension capabilities (the second example), and spatial understanding skills (the third example). Figure 15 illustrates that our **ROSS** is good at recognizing various visual

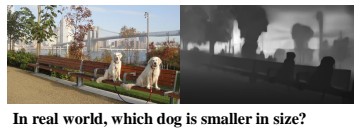 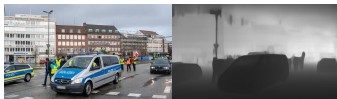 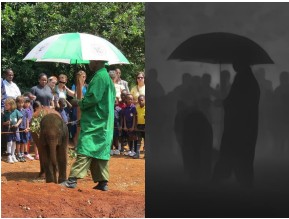

**In real world, which dog is smaller in size?**
**A. The dog closer to the camera.**
**B. The dog further to the camera.**
**C. They seem to be equally large.**
**D. It can not be decided from the image because information given is not enough.**

| LLaVA | A |
| LLaVA (w/ MiDaS) | A |
| **ROSS** | **B** |

**What is the positional relationship between the group of people with the flag and the black car?**
**A. Behind the black car.**
**B. Left of the black car.**
**C. Right of the black car.**
**D. In front of the black car.**

| LLaVA | D |
| LLaVA (w/ MiDaS) | A |
| **ROSS** | **D** |

**Has the man touched the elephant?**

| LLaVA | Yes. |
| LLaVA (w/ MiDaS) | Yes. |
| **ROSS** | **No.** |

Figure 16: **Qualitative comparisons** on SpatialBench (Cai et al., 2024). We take RGB + D inputs when testing. Notably, the extra depth expert MiDaS-3.0 (Birkl et al., 2023) sometimes *harms* comprehension (see the second example).

patterns, implying that the introduced reconstructive vision-centric objective indeed makes up the visual shortcomings of the original visual encoder.

Figure 16 provides qualitative results on SpatialBench (Cai et al., 2024). The extra depth understanding visual expert, *i.e.*, MiDaS (Birkl et al., 2023), fails to help LMMs understand depth maps both quantitatively in Table 5 and qualitatively in Figure 16.

# D    DISCUSSION

One limitation is that **ROSS** does not have generation capabilities, since **ROSS** is designed for enhanced multimodal comprehension, without the need to generate photorealistic aesthetic images. Furthermore, the gap in *training data* between comprehension and generation methods also matters. For instance, PixArt-$\alpha$ (Chen et al., 2023a), which is one of the most *efficient* text-to-image models, was trained on nearly 400M images to model the pixel discrepancy just in the *first* training stage. By contrast, our **ROSS** is only trained on nearly 3M images for one epoch. Future topics include achieving photorealistic text-to-image generation via incorporating more training samples.

