# OpenReview forum: "Reconstructive Visual Instruction Tuning"
_ICLR.cc/2025/Conference — ICLR 2025 Poster_

### Official Review · Reviewer_rXNr · 2024-10-21

**Soundness:** 2
**Presentation:** 3
**Contribution:** 2
**Rating:** 5
**Confidence:** 4

**Summary:**

This paper proposes a new variant of visual instruction tuning. Different from previous works that only utilize textual supervision, the proposed method additionally exploits visual supervision by reconstructing the contexts of input images. In particular, a denoising structure is introduced to better learn the latent representations. Experiments are conducted on several benchmarks.

**Strengths:**

1. The paper is well-organized and easy to read.
2. The motivation is straightforward.

**Weaknesses:**

1. The novelty is somewhat incremental. Utilizing image supervision is a very straightforward idea, and can be implemented by various ways. Denoising-based reconstruction is a very well-developed strategy in the field of image diffusion/generation. There is no specific in-depth design in the proposed architecture.

2. The authors claim that the proposed method capitalizes on the inherent richness and detail present within input images themselves, which are often lost in pure text supervision. In which cases, the information within the image is important? The authors should provide a more detailed analysis of this aspect. Is this information crucial for all common cases?

3. No experiments on complexity and efficiency. Of course, utilizing more self-supervision loss can definitely improve the model's performance. However, this image-aware training may cost more time and GPU memories compared to previous text-only supervision. The authors should provide an in-depth analysis of complexity and efficiency for fair comparison.

4. Given an image-text pair input, just some of the image-based contents are aligned with the text. I do not see any text-guided image reconstruction design in the architecture. This may help reduce the redundancy contexts.

**Questions:**

1. The novelty is somewhat incremental. Utilizing image supervision is a very straightforward idea, and can be implemented by various ways. Denoising-based reconstruction is a very well-developed strategy in the field of image diffusion/generation. There is no specific in-depth design in the proposed architecture.

2. The authors claim that the proposed method capitalizes on the inherent richness and detail present within input images themselves, which are often lost in pure text supervision. In which cases, the information within the image is important? The authors should provide a more detailed analysis of this aspect. Is this information crucial for all common cases?

3. No experiments on complexity and efficiency. Of course, utilizing more self-supervision loss can definitely improve the model's performance. However, this image-aware training may cost more time and GPU memories compared to previous text-only supervision. The authors should provide an in-depth analysis of complexity and efficiency for fair comparison.

4. Given an image-text pair input, just some of the image-based contents are aligned with the text. I do not see any text-guided image reconstruction design in the architecture. This may help reduce the redundancy contexts.

---

> ### Author Response · Authors · 2024-11-20
> **Rebuttal by Authors (Part 1)**
>
> We thank reviewer rXNr for the valuable time and constructive feedback. We appreciate your
> comments on the paper being “well-organized and easy to read” and the “straightforward” motivation.
> In the following, we have done our best to address your suggestions point-by-point.
>
> **W1: About the Novelty.**
>
> **A1:** We would like to emphasize that our main contribution is proposing a novel vision-centric
> supervision for enhanced comprehension capabilities for LMMs, instead of any specific technical
> modules. The underlying motivation is input images themselves inherently provide rich and detailed
> information, which is quite important for fine-grained comprehension tasks. As a result, we regard
> LMMs reconstruct input images as the supervision of those visual outputs.
>
> Actually, Reviewer kvNb uses “novel image-based supervision” to describe our work. Reviewer
> MCFE regards our work as a “step forward”. Reviewer SsLG says this paper is “very novel”, and
> Reviewer tGMa thinks our work is “inspiring”.
>
> (1) *The idea is not trivial.* While utilizing image supervision may *seem* straightforward, effectively
> using images to produce meaningful feedback through reconstruction for LMMs remains largely
> unexplored. *The key challenge lies in handling the heavy spatial redundancy of natural visual signals.*
> To address this, we systematically explore various reconstruction *targets* and *objectives*, which are
> definitely specific in-depth designs aimed at enhancing the comprehension capabilities of LMMs.
> Moreover, the self-attention module of the denoiser is specifically designed to manage the causal
> dependencies in the original visual outputs, ensuring that the model can effectively process and
> understand the whole visual content.
>
> (2) *Our contribution is not a newly introduced denoising strategy.* While denoising is a well-developed
> strategy in the field of image generation, we are the first to leverage the denoising objective as a
> reconstruction method *to boost fine-grained comprehension for LMMs.* Furthermore, *denoising is
> just one type of objective under our framework.* We adopt denoising simply because it alleviates the
> redundancy of natural visual signals. In fact, as demonstrated in Figure 7, vanilla regression still
> brings significant improvements, highlighting the flexibility and effectiveness of our approach.
>
> **W2: When Does the Inherent Details Become Important?**
>
> **A2:** We extend ablations to more representative benchmarks, where Ross manages to bring improvements in most cases. By systematically analyzing the results, we find the improvements are more
> significant on *fine-grained comprehension benchmarks such as HallusionBench, MMVP, ChartQA,
> and OCRBench*, as visual contents for these benchmarks are more crucial. In contrast, observed
> by Cambrian-1, LMMs can sometimes correctly answer the question without providing the
> image on knowledge benchmarks such as MMMU and AI2D, where the improvements seem to be
> less significant.
>
> |Benchmark|CLIP||||SigLIP||||
> |-|-|-|-|-|-|-|-|-|
> |LLM|Vicuna||Qwen2||Vicuna||Qwen2||
> ||LLaVA|Ross|LLaVA|Ross|LLaVA|Ross|LLaVA|Ross|
> |POPE-acc|86.3|**87.2 ↑ 0.9**|87.9|**88.4 ↑ 0.5**|86.0|**87.7 ↑ 1.7**|88.5|**88.7 ↑ 0.2**|
> |HallusionBench-aAcc|52.5|**55.8 ↑ 3.3**|55.0|**59.1 ↑ 4.1**|50.4|**53.8 ↑ 3.4**|57.3|**58.2 ↑ 0.9**|
> |MMBench-EN-dev|67.0|**67.6 ↑ 0.6**|73.8|**75.2 ↑ 1.4**|64.5|**69.2 ↑ 4.7**|76.3|**76.9 ↑ 0.6**|
> |MMBench-CN-dev|**60.0**|59.8 ↓ 0.2|72.9|**73.7 ↑ 0.8**|63.1|**63.4 ↑ 0.3**|75.7|**76.3 ↑ 0.7**|
> |SEED-img|**66.7**|66.4 ↓ 0.3|70.3|**70.7 ↑ 0.4**|68.2|**69.0 ↑ 0.8**|**72.3**|72.1 ↓ 0.2|
> |MMMU-dev|30.0|**34.0 ↑ 4.0**|44.0|**45.3 ↑ 1.3**|33.3|**38.0 ↑ 4.7**|38.7|**41.3 ↑ 2.6**|
> |MMMU-val|35.3|**36.0 ↑ 0.7**|41.9|**42.6 ↑ 0.7**|34.2|**35.4 ↑ 1.2**|41.8|**43.8 ↑ 2.0**|
> |MMVP|28.0|**36.3 ↑ 8.3**|29.6|**42.2 ↑ 12.6**|27.3|**38.0 ↑ 10.7**|40.7|**49.3 ↑ 8.6**|
> | AI2D-test | 61.2 | **61.4 ↑ 0.2** | 71.9 | **73.3 ↑ 1.4** | **62.6** | 62.4 ↓ 0.2|74.0|**74.5 ↑ 0.5**|
> | ChartQA-test | 32.9 | **39.8 ↑ 6.9** | 36.2 | **41.6 ↑ 5.4** | 34.0 | **48.2 ↑ 14.2** | 44.4 | **46.9 ↑ 2.5** |
> | DocVQA-val | 33.4 | **41.6 ↑ 8.2** | 31.1 | **44.7 ↑ 13.6** | 40.4 | **40.7 ↑ 0.3** | 39.2 | **39.3 ↑ 0.1** |
> | InfoVQA-val | 21.2 | **26.4 ↑ 5.2** | 22.1 | **39.3 ↑ 16.2** | 22.8 | **23.3 ↑ 0.5** | 24.0 | **25.1 ↑ 1.1** |
> | TextVQA-val | 55.7 | **58.7 ↑ 3.0** | 52.0 | **54.1 ↑ 2.1** | 60.5 | **62.6 ↑ 2.1** | 56.3 | **57.5 ↑ 1.2** |
> | OCRBench | 339 | **350 ↑ 11** | 363 | **381 ↑ 18** | 354 | **365 ↑ 11** | 432 | **448 ↑ 16** |
> | RealWorldQA | 52.7 | **53.2 ↑ 0.5** | 56.7 | **57.4 ↑ 0.7** | 55.0 | **57.1 ↑ 2.1** | 57.9 | **59.1 ↑ 1.2** |
> | **Average** | 47.8 | **50.6 ↑ 2.8** | 52.1 | **56.4 ↑ 4.3** | 49.2 | **52.4 ↑ 3.2** | 55.4 | **56.9 ↑ 1.5** |
>
> Examples in Figure 15 are vivid illustrations. Taking a close look at the given image becomes
> important to answer the question. Under such cases, vision-centric supervision enables LMMs to pay
> more attention to visual content, thereby enhancing overall performance.

---

> > ### Author Response · Authors · 2024-11-20
> > **Rebuttal by Authors (Part 2)**
> >
> > **W3: About Complexity and Efficiency.**
> >
> > **A3:** We apologize for the oversight in quantifying and discussing the computational overhead. To
> > address this concern, we have conducted additional experiments to measure the computational costs
> > and provide a clearer understanding of the practical trade-offs. Evaluations are conducted using 8
> > A100 GPUs with a global batch size of 128. Due to the limited GPU memory, we accumulate 4
> > gradient steps and the batch size per GPU is 4. The whole stage requires 5757 training steps. GPU
> > memories are averaged over 8 GPUs with DeepSpeed Zero 3. As demonstrated in the following table,
> > Ross introduces a negligible increase in training time and GPU memory.
> >
> > | Vision      | Base LLM             | $\mathcal{L}_{\mathrm{LMM}}^{\mathrm{visual}}$ | Trainable Parameters | Speed (s/iter) | Time       | GPU Memory  |
> > |-------------|----------------------|-----------------------------------------------|---------------------|----------------|------------|-------------|
> > | CLIP-L/336  | Qwen2-7B-Instruct    | --                                            | 7.63 B              | 8.31           | 13h 17min  | 45.34 G     |
> > | CLIP-L/336  | Qwen2-7B-Instruct    | ✔                                             | 7.68 B              | 9.02 (1.09 ×)  | 14h 25min  | 46.62 G (1.03 ×) |
> > | CLIP-L/336  | Vicuna-13B-v1.5      | --                                            | 13.05 B             | 13.33          | 21h 19min  | 48.62 G     |
> > | CLIP-L/336  | Vicuna-13B-v1.5      | ✔                                             | 13.11 B             | 14.69 (1.10 ×) | 23h 30min  | 49.07 G (1.01 ×) |
> > | SigLIP-L/384| Qwen2-7B-Instruct    | --                                            | 7.63 B              | 8.77           | 14h 1min   | 47.08 G     |
> > | SigLIP-L/384| Qwen2-7B-Instruct    | ✔                                             | 7.68 B              | 9.48 (1.08 ×)  | 15h 9min   | 52.07 G (1.11 ×) |
> > | SigLIP-L/384| Vicuna-13B-v1.5      | --                                            | 13.05 B             | 14.22          | 22h 44min  | 48.80 G     |
> > | SigLIP-L/384| Vicuna-13B-v1.5      | ✔                                             | 13.11 B             | 15.32 (1.08 ×) | 24h 30min  | 52.68 G (1.08 ×)
> >
> > **W4: About the Image-Text Alignment.**
> >
> > **A4:** We understand the reviewer’s concern regarding the alignment of image-based content with the
> > text. In fact, our Ross performs *vanilla reconstruction* instead of text-guided reconstruction. This is
> > because the visual tokens are always processed before the text instructions (as shown in Figure 2),
> > and the causal nature of LLMs means visual tokens do *not* interact with text inputs.
> >
> > We would like to clarify that the reconstructive pretext task does *not* aim for an enhanced image-text
> > alignment directly. Instead, its primary goal is to **mine the inherent information in the input images
> > that might be overlooked by sparse text instructions**. By reconstructing the images, the model can
> > extract more comprehensive visual features, providing a richer context that allows the LMMs to
> > decide which aspects to focus on based on the subsequent text instructions. As a result, this approach
> > contributes to better fine-grained comprehension of the full contents of the input images.

---

> ### Comment · Reviewer_rXNr · 2024-11-22
> **Reply to Authors' Response**
>
> I have carefully read the authors' responses. However, my main concerns still remain:
>
> 1. About the novelty of the developed method. The proposed framework is based on mature technologies (image-based supervision, reconstruction framework) from other fields. There is no discussion about it with existing technologies. Although authors claim that utilizing image contexts has not been explored in LMMs and their contribution is the entire system, I do not think the novelty of the idea meats a standard bar. The core technical designs are not new.
>
> 2. About the complexity comparison. The provided table seems not fair. The comparison is not apple-to-apple. Compared to LLMs, the vision encoder introduces much larger resource costs. If the authors want to compare their framework with existing LMMs, please provide the comparison with other LMM models like LLaVA, MiniGPT4.

---

> ### Author Response · Authors · 2024-11-22
>
> We sincerely thank the reviewer for their prompt and thoughtful feedback.
>
> **1. We respectfully disagree with their concerns regarding the novelty of our work.**
>
> First, we would like to clarify that the main contribution of our study lies in introducing a novel **vision-centric supervision approach** to enhance the comprehension capabilities of large multimodal models, rather than focusing on specific technical modules.
>
> Second, while we acknowledge the use of established techniques, we emphasize that the integration and demonstrated **effectiveness** of image-based reconstructive supervision within the context of LMMs constitute a significant and novel contribution. To the best of our knowledge, *no prior research has successfully employed "simple" image-based reconstructive supervision to improve comprehension in LMMs*. This challenge arises from the unique difficulty of generating meaningful low-level visual supervision for highly semantic models like LMMs.
>
> This work conducted a systematic study on (1) targets and (2) objectives, with the ultimate goal of *handling the heavy spatial redundancy of natural visual signals*. Specifically,
>
> - **Towards reconstruction targets**, we study reconstructing (i) vanilla RGB pixel values, (ii) latent tokens obtained by deep models, and (iii) RGB pixels using a pixel decoder.
> - **Towards reconstruction objectives**, we empirically found that denoising is more suitable than vanilla regression for the ultimate goal as it avoids fitting specific values.
>
> In fact, *very few studies have begun to focus on the design of visual supervision for LMMs*.
>
> Therefore, **our work represents a pioneering step forward**, providing a strong baseline for adding visual supervision to LMMs and improving fine-grained comprehension capabilities. We hope our findings will inspire future research and innovations in this area.
>
> We believe that the systematic exploration and integration of these components into a cohesive framework constitute a significant contribution to the field.
>
> **2. We would like to clarify that our complexity comparison is indeed fair and apples-to-apples.**
>
> We keep all unrelated factors, including the vision encoder, the language model (LLM), and the training data, *identical* across each two rows in the table.
> The **only difference** between the two compared methods is the incorporation of our proposed objective.
> Therefore, we maintain that **the provided complexity comparison is completely fair and apples-to-apples.**
>
> Following your suggestions, we estimated the computational costs **under the exact same setting as the *original* LLaVA-v1.5** in the following table. Specifically,
> - The visual encoder is set to CLIP-ViT-L/14@336 and is kept frozen.
> - The LLM is either Vicuna-7B-v1.5 or Vicuna-13B-v1.5.
> - The training data is LLaVA-665K, where the training requires 5197 steps with a global batch size of 128.
>
> *The only difference is our Ross incorporates $\mathcal{L}_{\mathrm{LMM}}^{\mathrm{visual}}$ while LLaVA-v1.5 does not.*
> This ensures that any observed differences in complexity are directly attributable to the inclusion of our proposed loss function.
>
> |Method|Vision|LLM|$\mathcal{L}_{\mathrm{LMM}}^{\mathrm{visual}}$|Trainable Parameters|Speed|Time|
> |-|-|-|-|-|-|-|
> |LLaVA-v1.5-7B|CLIP|Vicuna-7B-v1.5|--|6.76 B|6.84|9h 52min|
> |Ross-7B|CLIP|Vicuna-7B-v1.5|✔|6.81 B|7.58 (1.11×)|10h 56min|
> |LLaVA-v1.5-13B|CLIP| Vicuna-13B-v1.5|--|13.05 B|13.33|19h 15min|
> |Ross-13B|CLIP|Vicuna-13B-v1.5|✔|13.11 B|14.69 (1.10×)|21h 12min|
>
> It was officially reported in the LLaVA's GitHub repo [1] that, using DeepSpeed ZeRO-3 on 8xA100, it takes approximately 20 hours for LLaVA-v1.5-13B and around 10 hours for LLaVA-v1.5-7B. Our implementation, under the same conditions, yields similar training times, confirming the reliability of our setup.
>
> With regard to MiniGPT-4, it is hard to fairly compare its computational cost with that of LLaVA (-v1.5) or Ross due to substantial differences in model settings. Specifically:
> - **Training data:** LLaVA-v1.5 utilized 558K and 665K samples for pre-training and instruction tuning, respectively. MiniGPT4 incorporated 5M samples for training.
> - **Training recipe:** LLaVA-v1.5 adopts a two-stage training pipeline, where the first stage is for training the projector, while the second stage is for training both the projector and the LLM. MiniGPT4 has only one training stage for the projector.
> - **Visual encoder:** LLaVA-v1.5 utilized CLIP-L-336 (0.3 B), while MiniGPT4 adopted EVA-CLIP-G-224 (1B).
>
> Given these differences, a fair comparison with MiniGPT-4 is not feasible.
>
> However, the core idea behind Ross, namely **vision-centric supervision**, represents a general enhancement for visual instruction tuning. We believe this approach could also be applied to MiniGPT-4-like models with minimal computational overhead, as evidenced by our experiments on LLaVA-based models.
>
> **References**
>
> [1] https://github.com/haotian-liu/LLaVA

---

> ### Comment · Reviewer_rXNr · 2024-11-25
> **Reply to Authors' New Response**
>
> Thanks for the authors' further clarification. However, my concerns about the paper's novelty remain.
>
> I indeed recognize the motivation and the "step forward" of this work. However, utilizing existing technologies to address a new setting is not novel. The authors also admit that they just integrate established techniques to develop a simple supervision framework. The designs of reconstruction targets and objectives are not new and exciting.
>
> An interesting high-level idea does not mean that it is novel enough for a high-quality conference. The core technology is also important in supporting its detailed designs with novel inspiration.
>
> Therefore, I believe this paper does not meet the bar for ICLR.

---

> > ### Author Response · Authors · 2024-11-25
> >
> > Thank you for recognizing the motivation and the "step forward" of this work.
> >
> > While we respect your perspective on novelty, we believe that *an interesting and effective high-level idea does represent a significant contribution*.
> > We agree that there is room for further improvement, and in our future work, we will focus on enhancing the denoiser and establishing specific modules to achieve better performance.
> >
> > Thank you again for your prompt and valuable feedback.

---

### Official Review · Reviewer_tGMa · 2024-10-31

**Soundness:** 3
**Presentation:** 3
**Contribution:** 3
**Rating:** 6
**Confidence:** 4

**Summary:**

This paper introduces Reconstructive Visual Instruction Tuning (ROSS), a novel approach that leverages input images as additional supervision signals to enhance fine-grained visual perception capabilities. Through extensive empirical studies, the authors explore optimal training settings, such as auxiliary module design and the nature of supervision signals. Experimental results indicate that ROSS consistently improves the performance of existing vision-language models.

**Strengths:**

1. The paper is well-organized and easy to follow, with a clear presentation of the key ideas. It is always inspiring to see that a simple auxiliary loss can improve the performance of large multimodal models.
2. The authors have put a lot of effort into designing experiments that thoroughly ablate the proposed training methods. This rigor in experimentation is highly appreciated and adds to the paper’s credibility.

**Weaknesses:**

1. The proposed method lacks novelty and insight. As noted in Section 2, previous work, such as [1], has explored using images as additional supervision in the form of generative objectives. Consequently, the contribution of this paper is limited, focusing mainly on a variation of established methods.
2. The proposed method lack interpretability. While using pixel-level reconstruction as an auxiliary task may enhance fine-grained image recognition, this approach risks impairing the language capabilities of the multimodal model, as the final task output is high-level semantic language. The authors provide insufficient experiments and explanations regarding this trade-off, leaving questions about potential impacts on language performance.
3. The scalability of the empirical findings is uncertain. It remains unclear whether the optimal settings identified would hold in different scenarios or with variations in model scale, training data, and other factors. Although the authors attempt to address this concern with results in Table 3, these efforts are insufficient, as many relevant variables remain unexplored.

[1] Sun Q, Cui Y, Zhang X, et al. Generative multimodal models are in-context learners[C]//Proceedings of the IEEE/CVF Conference on Computer Vision and Pattern Recognition. 2024: 14398-14409.

**Questions:**

1. The results in Table 4 appear counter-intuitive, as ROSS-13B performs consistently worse than ROSS-7B. This raises concerns about whether the proposed method is well-suited to larger-scale models. Clarification on this disparity and potential scalability issues would strengthen the paper.
2. The analysis in Section 5.2 is unclear. My understanding is that the authors aim to demonstrate that additional visual supervision enables the model to better focus on relevant areas of the image during VQA tasks. However, the reasoning behind this effect is not well-explained. Further elaboration on the mechanisms or evidence supporting this claim would enhance interpretability.

---

> ### Author Response · Authors · 2024-11-20
> **Rebuttal by Authors (Part 1)**
>
> We thank reviewer tGMa for the valuable time and constructive feedback. We are particularly
> grateful for your kind words about the paper being "well-organized and easy to follow" with a
> "clear presentation of the key ideas." Your observation that "a simple auxiliary loss can improve the
> performance of LMMs" is very inspiring and aligns with our goals. We have done our best to address
> your suggestions point-by-point in our response below.
>
> **W1: About Novelty and Insight.**
>
> **A1:** We would like to clarify that **"generation" and "reconstruction" are fundamentally different
> approaches.** Previous works (Emu-2) explore “generative objectives”, while our Ross is a kind
> of “reconstructive objective”. The detailed pipeline comparison can be found in Figure 11 in the
> Appendix. Specifically, Emu-2 takes outputs corresponding to *learnable queries* as conditions, while
> our Ross takes *outputs corresponding to visual inputs*.
>
> Empirically, our *Ross significantly outperforms Emu-2 in comprehension tasks*, demonstrating the
> effectiveness of “reconstruction” over “generation”. Moreover, we have compared reconstruction and
> generation in Table 2, where *using generative objectives actually fails to bring improvements* over the
> baseline. This empirical evidence highlights the effectiveness of the reconstructive approach over the
> generative one.
>
> Both the distinct pipeline and the superior performance of Ross underscore our contributions and
> insights.
>
> **W2: Pixel-Level Reconstruction may Risk the Langauge Capabilities.**
>
> **A2:** Following suggestions, we evaluate multi-modal benchmarks that mainly require general knowledge following Cambrian-1, including ScienceQA, MMMU, and AI2D. Furthermore, we incorporate
> representative language benchmarks, including general understanding on MMLU and HellaSwag,
> and instruction-following on IFEval. Empirical results demonstrate that *Ross does not harm language
> capabilities as it brings improvements in most cases.*
>
> |Benchmark|CLIP||||SigLIP||||
> |-|-|-|-|-|-|-|-|-|
> |LLM|Vicuna||Qwen2||Vicuna||Qwen2|||
> ||LLaVA|Ross|LLaVA|Ross|LLaVA|Ross|LLaVA|Ross|
> | ScienceQA-test | 68.5 | **69.0 ↑ 0.5** | 76.5 | **77.4 ↑ 0.9** | 69.6 | **71.3 ↑ 1.7** | 78.3 | **78.5 ↑ 0.2** |
> | MMMU-dev | 30.0 | **34.0 ↑ 4.0** | 44.0 | **45.3 ↑ 1.3** | 33.3 | **38.0 ↑ 4.7** | 38.7 | **41.3 ↑ 2.6** |
> | MMMU-val | 35.3 | **36.0 ↑ 0.7** | 41.9 | **42.6 ↑ 0.7** | 34.2 | **35.4 ↑ 1.2** | 41.8 | **43.8 ↑ 2.0** |
> | AI2D-test | 61.2 | **61.4 ↑ 0.2** | 71.9 | **73.3 ↑ 1.4** | **62.6** | 62.4 ↓ 0.2 | 74.0 | **74.5 ↑ 0.5** |
> | MMLU | 26.5 | **27.4 ↑ 0.9** | 57.1 | **60.7 ↑ 3.6** | **26.0** | 25.9 ↓ 0.1 | 60.9 | **61.0 ↑ 0.1** |
> | HellaSwag-acc-norm | **27.0** | 26.9 ↓ 0.1 | **46.4** | 46.2 ↓ 0.2 | **27.1** | 27.0 ↓ 0.1 | 45.5 | **46.6 ↑ 1.1** |
> | IFEval-strict-inst | 41.2 | **44.6 ↑ 3.4** | 47.1 | **49.2 ↑ 2.1** | 43.6 | **43.8 ↑ 0.2** | 47.8 | **48.1 ↑ 0.3** |
> | IFEval-strict-prompt | 28.7 | **35.3 ↑ 6.7** | 35.1 | **37.0 ↑ 1.9** | 32.5 | **33.1 ↑ 0.6** | 35.3 | **36.2 ↑ 0.9** |
> | **Average** | 39.8 | **41.8 ↑ 2.0** | 52.5 | **54.0 ↑ 1.5** | 41.1 | **42.1 ↑ 1.0** | 52.8 | **53.8 ↑ 1.0** |
>
> **W3: Scability.**
>
> **A3:**
> We appreciate the reviewer’s insightful feedback regarding the scalability of our empirical
> findings. To address these concerns, we have conducted additional experiments to study (1) the model
> scaling behavior and (2) the data scaling behavior of Ross.
>
> (1) To study the stability and scalability of Ross across different model sizes, we use the Qwen2.5
> series with varying sizes as the base language model while keeping the CLIP-ViT-L/14@336 as the
> visual encoder. The pre-training data is LLaVA-558K, and the instruction tuning data is LLaVA-665K.
> The results, shown in the following table, demonstrate that *Ross consistently brings improvements
> over the baseline (LLaVA) across different model sizes.*
>
> |Benchmark|0.5B||1.5B||3B||7B||
> |-|-|-|-|-|-|-|-|-|
> ||LLaVA|Ross|LLaVA|Ross|LLaVA|Ross|LLaVA|Ross|
> |POPE-acc|50.0|**60.4 ↑ 10.4**|85.3|**87.9 ↑ 2.4**|87.3|**88.1 ↑ 0.8**|87.9|**88.4 ↑ 0.5**|
> |HallusionBench-aAcc|45.8|**48.0 ↑ 2.2**|48.7|**49.6 ↑ 0.9**|52.2|52.2 – 0.0|48.7|**53.7 ↑ 5.0**|
> |MMBench-EN-dev|55.2|**60.4 ↑ 5.2**|67.5|**68.2 ↑ 1.7**|70.6|**71.4 ↑ 0.8**|75.0|**75.7 ↑ 0.7**|
> |MMBench-CN-dev|45.6|**48.9 ↑ 3.3**|62.4|**63.9 ↑ 1.5**|68.0|**69.1 ↑ 1.1**|**73.6**|73.5 ↓ 0.1|
> |SEED-img|**55.8**|55.6 ↓ 0.2|66.3|**66.8 ↑ 0.5**|68.2|**68.4 ↑ 0.2**|70.6|**71.0 ↑ 0.4**|
> |OCRBench|229|**248 ↑ 19**|291|**298 ↑ 7**|**313**|308 ↓ 5|334|**358 ↑ 24**|
> |MMMU-dev|35.2|**36.0 ↑ 0.8**|44.7|**45.0 ↑ 0.3**|48.7|**49.0 ↑ 0.3**|48.0|48.0 – 0.0|
> |MMMU-val|38.0|**40.3 ↑ 1.7**|41.8|**43.6 ↑ 1.8**|41.6|**42.7 ↑ 1.1**|47.3|**48.0 ↑ 0.7**|
> |AI2D-test|45.3|**46.0 ↑ 0.7**|59.0|**59.5 ↑ 0.5**|62.9|**63.2 ↑ 0.3**|68.3|**68.5 ↑ 0.2**|
> |RealWorldQA|45.1 | **46.4 ↑ 1.3** | 50.5| **53.5 ↑ 3.0**|55.7|**57.9 ↑ 2.2**|59.5|**59.9 ↑ 0.4**|
> |**Average**|43.9|**46.7 ↑ 2.8**|55.3|**56.8 ↑ 1.5**|58.9|**59.3 ↑ 0.4**|61.2|**62.3 ↑ 1.1**|
>
> (To be continued)

---

> > ### Author Response · Authors · 2024-11-20
> > **Rebuttal by Authors (Part 2)**
> >
> > (2) Next, to study the impact of the training data scale, we used Qwen2-7B-Instruct as the base
> > language model and CLIP-ViT-L/14@336 as the visual encoder. We compared the performance of
> > Ross and the baseline under different scales of training data. The following table demonstrates that
> > *Ross consistently brings significant improvements as the training data scale increases.*
> >
> > | PT   | SFT  | $\mathcal{L}_{\mathrm{LMM}}^{\mathrm{visual}}$ | POPE     | Hallu.  | ChartQA  | OCRBench | MMB$^{\text{EN}}$ | AI2D    |
> > |----|---|----|---|----|-----|---|--|-|
> > | 558K | 737K | -- | 87.9  | 55.0    | 34.0 | 363 | 73.8 | 72.4    |
> > | 558K | 737K | ✔ | **88.4 ↑ 0.5** | **59.1 ↑ 4.1** | **40.4 ↑ 6.4** | **380 ↑ 17** | **75.2 ↑ 1.4** | **73.3 ↑ 0.9** |
> > | 558K | 1.2M | --  | 88.5     | 57.3    | 37.0     | 389      | 75.7             | 74.5    |
> > | 558K | 1.2M | ✔   | **88.8 ↑ 0.3** | **57.8 ↑ 0.5** | **42.0 ↑ 5.0** | **392 ↑ 3** | **76.8 ↑ 1.1** | **74.7 ↑ 0.2** |
> > | 2M   | 737K | --  | 88.1     | 55.6    | 37.3     | 384      | 76.2             | 72.3    |
> > | 2M | 737K | ✔  | **88.3 ↑ 0.2** | **56.2 ↑ 0.6** | **41.9 ↑ 4.5** | **398 ↑ 14** | **77.0 ↑ 0.8** | **73.4 ↑ 1.1** |
> > | 2M   | 1.2M | -- | 88.5     | 53.8    | 41.2     | 388      | 76.5             | 73.9    |
> > | 2M | 1.2M | ✔ | **88.9 ↑ 0.4** | **57.3 ↑ 2.5** | **43.2 ↑ 2.0** | **405 ↑ 17** | **78.0 ↑ 1.5** | **74.1 ↑ 0.2** |
> >
> > **Q1: Ross-13B performs consistently worse than Ross-7B.**
> >
> > **A4:** We would like to clarify that the performance of LMMs largely depends on the base LLM. In
> > our experiments, Ross-13B is based on Vicuna-13B-v1.5, while Ross-7B is based on Qwen2-7B-
> > Instruct, which is a much stronger LLM backbone. This difference in base models can explain why
> > Ross-13B performs worse than Ross-7B. Similar issues have been observed in other methods. For
> > example, Cambrian-1-13B performs worse than Cambrian-1-8B in most cases because the former
> > uses Vicuna-13B-v1.5, while the latter uses Llama3-8B-Instruct.
> >
> > To further investigate this issue, we conducted additional experiments using Vicuna-v1.5 series as
> > the language model while keeping the training data the same, resulting in Ross-7B-vicuna and Ross-
> > 13B-vicuna, respectively. Empirical results demonstrate that Ross-13B-vicuna significantly outperforms
> > Ross-7B-vicuna. This indicates that Ross is indeed well-suited to larger-scale models when the base
> > language models are comparable.
> >
> > | Model | POPE | Hallu.  | MMBench-EN-dev | MMBench-CN-dev | SEED-img | MMMU| MMVP| GQA  | AI2D|
> > |-|-|-|-|-|-|-|-|-|-|
> > | *Base LLM: Vicuna-7B-v1.5* |
> > | LLaVA-v1.5-7B     | 86.2     | 47.5    | 65.5              | 58.5              | 66.0             | 34.4    | 20.0    | 62.0    | 55.4    |
> > | LLaVA-v1.6-7B     | 86.5     | 35.8    | 67.4              | 60.1              | **70.2**         | 35.8    | 37.3    | **64.2**| 67.1    |
> > | **Ross-7B-vicuna** | **88.2** | **55.2** | **67.7**          | **61.3**          | 67.6             | **36.9**| **39.3**| 63.7    | **69.3**|
> > | |
> > | *Base LLM: Vicuna-13B-v1.5* |
> > | LLaVA-v1.5-13B   | 82.5     | 44.9    | 68.8              | 63.6              | 68.2             | 36.6    | 31.9    | 63.3    | 60.8    |
> > | LLaVA-v1.6-13B | 86.2     | 36.7    | 70.0              | 64.1              | 71.9             | 36.2    | 35.6    | **65.4**| 72.4    |
> > | Mini-Gemini-13B          | --       | --      | 68.6              | --                | 73.2             | 37.3    | 19.3    | 63.7    | 70.1    |
> > | Cambrian-1-13B           | 85.7 | 54.0 | **75.7**          | 65.9      | **74.4**         | 40.0    | 41.3    | 64.3    | 73.6    |
> > | **Ross-13B-vicuna** | **88.7** | **56.4** | 73.6              | **67.4**          | 71.1             | **41.3**| **44.7**| 65.2    | **73.8**|
> >
> > **Q2: Further Elaborations.**
> >
> > **A5:** To better explain the reasoning behind how the vison-centric supervision enables the model
> > to focus on relevant areas of the image during VQA tasks, we provide a qualitative comparison
> > using GradCAM on MMVP, since GradCAM helps
> > in understanding which parts of the image the model is focusing on, making the model’s decision-
> > making the process more transparent. *Please refer to Figure 12 in Appendix C.2.* Specifically, it works
> > by computing the gradients of the target class with respect to the feature maps in a specific layer of
> > the network, typically the last convolutional layer for CNNs. These gradients are then weighted and
> > summed to produce a heat map that highlights the regions of the input image that are most important
> > for the prediction.
> >
> > In our analysis, we visualize the gradient of the second-to-last block of the LMM, regarding the option
> > of the ground-truth answer as the target class. Specifically in this case, where the providing question
> > is about the spider web, our proposed vision-centric supervision signals provide more reasonable
> > gradients and urge LMMs to focus on relevant regions, i.e., the spider web, as the training evolves.

---

> > > ### Comment · Reviewer_tGMa · 2024-11-23
> > > **Reply to Authors' Response**
> > >
> > > Thanks to the authors for their significant effort to provide more comprehensive results, which has addressed many of my initial concerns. I believe the empirical findings presented will make a valuable contribution to the community, particularly in the area of multi-modal large models. Consequently, I have decided to increase my review score to 6.
> > >
> > > However, I still believe that the proposed method lack sufficient interpretability as mentioned in W2. The approach of adding visual supervision to the output part of a language model and then testing it on a task like Visual Question Answering (VQA) remains counterintuitive. Despite reviewing the explanations provided in Section 5.2 and the appendix, I find that these do not fully address the underlying concerns. I would appreciate a training framework that is either more interpretable or intuitive. I look forward to further clarifications that could potentially enhance the robustness and understanding of the proposed methods.

---

> > > > ### Author Response · Authors · 2024-11-23
> > > >
> > > > We sincerely thank the reviewer for the prompt and constructive feedback.
> > > >
> > > > We understand the concern regarding the interpretability of our proposed method. The common problem when testing LMMs on VQA tasks is the *unconditional preference* problem [1]. That is, the model often *overlooks* the given image, and researchers have begun to pay attention to this phenomenon in preference optimization [1] and hallucination mitigation [2].
> > > >
> > > > The provided empirical explanations demonstrate that *supervising visual outputs* alleviates this issue both (1) consequently in the final results, and (2) progressively over the training procedure. Specifically,
> > > > - **Final Results:** As shown in Table 1, the attention scores of the ground-truth answer with respect to all visual tokens are significantly improved by incorporating our Ross. This indicates that LMMs focus more on the image content when answering the question.
> > > > - **Training Procedure:** Figure 12 illustrates that as training progresses, the gradient allows the model to focus on specific question-related image content. This progressive improvement highlights the effectiveness of our approach in enhancing the model's attention to visual information.
> > > >
> > > > The primary source of improvement in Ross is its increased focus on image content, and incorporating visual supervision reduces the possibility of overlooking the image.
> > > >
> > > > We have made efforts to enhance the interpretability of our Ross through both quantitative analysis (Table 1) and qualitative analysis (Figure 12). However, we are open to further improvements and would greatly appreciate specific instructions or suggestions from the reviewer on how to make the training framework more interpretable or intuitive.
> > > >
> > > >
> > > > **References**
> > > >
> > > > [1] Fei Wang, et al. mDPO: Conditional Preference Optimization for Multimodal Large Language Models. arXiv preprint arXiv:2406.11839, 2024.
> > > >
> > > > [2] Sicong Leng, et al. Mitigating object hallucinations in large vision-language models through visual contrastive decoding. CVPR, 2024.

---

### Official Review · Reviewer_SsLG · 2024-11-02

**Soundness:** 4
**Presentation:** 3
**Contribution:** 4
**Rating:** 6
**Confidence:** 4

**Summary:**

This paper introduces ROSS, a novel approach to enhance Large Multimodal Models (LMMs) through vision-centric supervision signals. Unlike conventional visual instruction tuning that only supervises text outputs, ROSS introduces a reconstructive objective where LMMs must reconstruct input images' latent representations.

The authors address the challenge of spatial redundancy in visual signals by employing a denoising objective to reconstruct latent representations rather than raw RGB values. The approach demonstrates significant improvements across various benchmarks, particularly in fine-grained visual comprehension and hallucination reduction, while maintaining a lightweight architecture compared to existing methods that rely on multiple visual experts.

**Strengths:**

This paper is very novel and address the very important topic on vision-centric learning in LMM.

Specifically, the paper introduces an innovative vision-centric supervision method that leverages the inherent richness of input images, addressing a clear gap in existing LMM training approaches. The use of denoising objectives for latent representation reconstruction is particularly clever as it handles the spatial redundancy problem.

The authors conduct extensive experiments across multiple benchmarks, including thorough ablation studies that systematically evaluate different components of their approach (regression vs. denoising, different tokenizers, architecture choices). The comparison with state-of-the-art methods is particularly thorough.

ROSS achieves competitive or superior performance using only a single visual encoder, making it more efficient than existing approaches that require multiple visual experts. This has significant practical implications for deployment and scalability.

The methodology is well-grounded in existing literature and builds thoughtfully on previous work in both vision and language domains. The authors clearly explain how their approach differs from both traditional visual instruction tuning and newer aggregated visual instruction tuning methods.

**Weaknesses:**

I think the major weakness is about the Computational Costs. While the paper emphasizes the efficiency of using a single visual encoder, it lacks detailed analysis of training time, memory requirements, and computational costs compared to baseline methods.

Besides, the paper doesn't thoroughly discuss the sensitivity of ROSS to various hyperparameters, such as the denoising schedule or architecture choices. It would be benefitical to add this part analysis and show ROSS's denoising part is robust and easy to train.

**Questions:**

This not be a major issue, but it's not very natural to see that using vision encoder to encode image pixels into LLM's embeddings and use another denosing module to reconstruct it back to image pixels.

This may introduce improvements since the model better maintains the information of original images, but it may be more natural to see it's used in a quantized tokenizer based LMM like EMU-3 and Chamelon.

I was wondering how the authors feel about this and have insights on this question?

---

> ### Author Response · Authors · 2024-11-20
> **Rebuttal by Authors (Part 1)**
>
> We thank Reviewer SsLG for the insightful and positive feedback. We are deeply appreciative of
> using "very novel" and the recognition of the importance of vision-centric learning in LMMs. The
> acknowledgment of our "innovative vision-centric supervision method" and the "clever" use of
> denoising objectives is highly encouraging. We are also grateful for your praise of our "extensive
> experiments" and "thorough ablation studies". We provide point-to-point responses below.
>
> **W1: Computational Costs.**
>
> **A1:**
> We apologize for the oversight in discussing the computational overhead quantitatively. To
> address this concern, we have conducted additional experiments in the following table to measure the
> computational costs and provide a clearer understanding of the practical trade-offs. Evaluations are
> conducted using 8 A100 GPUs with a global batch size of 128, where the batch size per GPU remains
> 4 4 gradient steps are accumulated. GPU memories are averaged over 8 GPUs with DeepSpeed Zero
> 3. As demonstrated in the following table, Ross introduces a marginal increase in training time and
> GPU memory.
>
> | Vision      | Base LLM             | $\mathcal{L}_{\mathrm{LMM}}^{\mathrm{visual}}$ | Trainable Parameters | Speed (s/iter) | Time       | GPU Memory  |
> |-------------|----------------------|-----------------------------------------------|---------------------|----------------|------------|-------------|
> | CLIP-L/336  | Qwen2-7B-Instruct    | --                                            | 7.63 B              | 8.31           | 13h 17min  | 45.34 G     |
> | CLIP-L/336  | Qwen2-7B-Instruct    | ✔                                             | 7.68 B              | 9.02 (1.09 ×)  | 14h 25min  | 46.62 G (1.03 ×) |
> | CLIP-L/336  | Vicuna-13B-v1.5      | --                                            | 13.05 B             | 13.33          | 21h 19min  | 48.62 G     |
> | CLIP-L/336  | Vicuna-13B-v1.5      | ✔                                             | 13.11 B             | 14.69 (1.10 ×) | 23h 30min  | 49.07 G (1.01 ×) |
> | SigLIP-L/384| Qwen2-7B-Instruct    | --                                            | 7.63 B              | 8.77           | 14h 1min   | 47.08 G     |
> | SigLIP-L/384| Qwen2-7B-Instruct    | ✔                                             | 7.68 B              | 9.48 (1.08 ×)  | 15h 9min   | 52.07 G (1.11 ×) |
> | SigLIP-L/384| Vicuna-13B-v1.5      | --                                            | 13.05 B             | 14.22          | 22h 44min  | 48.80 G     |
> | SigLIP-L/384| Vicuna-13B-v1.5      | ✔                                             | 13.11 B             | 15.32 (1.08 ×) | 24h 30min  | 52.68 G (1.08 ×) |
>
> **W2: Sensitivity to Hyperparameters.**
>
> **A2:**
> We appreciate the reviewer’s suggestion to thoroughly discuss the sensitivity of Ross. We
> study the effectiveness of different schedules of β in the following table, where all methods are
> equipped with CLIP-VIT-L/14@336 and Qwen2-7B-Instruct. The pre-training data is LLaVA-558K
> and the instruction tuning data is Cambrian-737K. From the table, we can tell that even with different
> schedules of β, Ross *consistently* improves the baseline, demonstrating its robustness to the denoising
> schedule.
>
> | Schedule of $\beta$          | POPE     | HallusionBench  | MMVP    | ChartQA  | MMBench-EN-dev |
> |------------------------------|----------|---------|---------|----------|------------------|
> | --                           | 87.9     | 55.0    | 29.6    | 34.0     | 73.8             |
> | Linear [R1] | 88.1 ↑ 0.2 | 57.3 ↑ 2.3 | 42.0 ↑ 12.4 | 39.2 ↑ 5.2 | 75.1 ↑ 1.3 |
> | Scaled Linear [R2] | **88.4 ↑ 0.5** | 58.3 ↑ 3.3 | 40.0 ↑ 10.4 | **40.7 ↑ 6.7** | 75.3 ↑ 1.5 |
> | GLIDE Softmax [R3] | **88.4 ↑ 0.5** | **59.1 ↑ 4.1** | **42.2 ↑ 12.6** | 40.4 ↑ 6.4 | 75.2 ↑ 1.4 |
> | GeoDiff Sigmoid [R4] | 88.2 ↑ 0.3 | 57.7 ↑ 2.7 | 41.3 ↑ 11.7 | 38.9 ↑ 4.9 | **75.5 ↑ 1.7** |
>
> As for the architecture choices, we have analyzed the impact of different visual tokenizers. The
> results, presented in Figure 10, show that the KL-16 tokenizer outperforms the VQ-16 tokenizer.
> One intuitive explanation is that KL-16 preserves more low-level details compared to VQ-16, as
> quantization can lead to information loss. Additionally, Figure 10 highlights the importance of the
> self-attention module. Since the original visual outputs are causal, modeling inter-token dependencies
> via self-attention is crucial. The number of trainable parameters for the denoiser is not the primary
> factor affecting performance.
>
> **References**
>
> [R1] Jonathan Ho, et al. Denoising diffusion probabilistic models. NeurIPS, 2020.
>
> [R2] Robin Rombach, et al. High-resolution image synthesis with latent diffusion models. CVPR, 2022.
>
> [R3] Alexander Quinn Nichol, et al. Glide: Towards photorealistic image generation and editing with text-guided diffusion models. ICML, 2022.
>
> [R4] Minkai Xu, et al. Geodiff: A geometric diffusion model for molecular conformation generation. ICLR, 2022

---

> > ### Author Response · Authors · 2024-11-20
> > **Rebuttal by Authors (Part 2)**
> >
> > **Q1: Comparison with VQ-based LMMs.**
> >
> > **A3:** The main reasons for not following VQ-based methods such as Emu-3 and Chameleon are the
> > training efficiency and data requirements. They often require extensive training with large amounts
> > of caption data to achieve robust image-text alignment. Despite this, their comprehension capabilities
> > can still lag behind (Chameleon) or just be comparable (Emu-3) to the LLaVA baseline. In contrast,
> > LLaVA's plug-in architecture is much more data-efficient. This efficiency is crucial for practical
> > applications with reasonable GPU requirements. Therefore, we mostly follow LLaVA's settings and leverage a denoiser to recover the high-level LMM’s features back to the pixel space, which may be a
> > little bit unconventional.
> >
> > The underlying insight driving our design is that **high-level features from LMMs can be mapped
> > into the pixel space.** To support this claim, we fine-tune the denoiser to recover latent tokens from
> > a frozen KL-16 conditioned on Ross-7B features on ImageNet-1K for only five epochs, where the
> > denoiser manages to produce reasonable reconstruction results (*please refer to Figure 9 in the revised
> > manuscript*). This interesting finding demonstrates that *Ross-7B features actually contain image
> > details*. However, the two-layer MLP adopted by LLaVA may be insufficient to fully extract this
> > inherent information, hence the need for an extra denoising module.

---

> > > ### Author Response · Authors · 2024-11-27
> > >
> > > Dear Reviewer SsLG,
> > >
> > > As the ICLR discussion phase is nearing its conclusion, we are writing to kindly ask that you review our responses to the comments and questions raised during the review process. Your thorough examination and any additional feedback or discussions you may wish to initiate will be crucial in refining our work. We look forward to your final ratings and any further dialogue that may enhance our paper.
> > >
> > > Sincerely,
> > >
> > > Authors

---

> > > > ### Author Response · Authors · 2024-12-02
> > > >
> > > > Thank you again for your insightful and positive feedback. We believe our rebuttal has addressed your questions and concerns. With the discussion phase deadline approaching, we would greatly appreciate it if you could let us know if you have any additional questions. We are happy to respond as soon as possible.

---

### Official Review · Reviewer_MCFE · 2024-11-04

**Soundness:** 3
**Presentation:** 3
**Contribution:** 3
**Rating:** 6
**Confidence:** 4

**Summary:**

The paper uses image denoising as an auxiliary training task to improve VLMs abilities.
Denoising encourages the VLM to preserve image detail.
The work is motivated by the MAE (masked auto-encoder) line of work for training foundational vision encoders.
The auxiliary task helps VLMs achieve higher benchmark numbers.

**Strengths:**

Whereas text-based LLMs have achieved amazing results only with next-token prediction, when we have image + text VLMs, it has always seemed that only doing next-token prediction for text could be improved upon.  In that regard, the technique proposed in this paper, to use image denoising as a pretext task, seems like step forward, as a way to add more supervision to the VLM and to improve results.

The benefits to the metrics are actually significant in some cases, not just epsilon levels, which is great to see.

It seems to me like the method is described clearly and the results are presented clearly.

**Weaknesses:**

1. I wish the benchmarks cited in the paper to measure the benefits of their method, i wish those benchmarks more closely matched recent popular work such as "The Llama 3 Herd of Models" or "Qwen2-VL", which include benchmarks like TextVQA, DocVQA, etc ... It may not change the conclusion but when we compare methods, it's important to look at a representative distribution of benchmarks. Table 4 has some of these common benchmarks, but not all of them. Furthermore, i wish Table 4 (or perhaps Table 3) included the same benchmarks but also had a very clean A/B experiment that was a baseline without the method vs. using the method. Table 3 has this but Table 3 has a different set of benchmarks! So it's rather confusing what conclusion to draw.

2. The fact that the work was done at lower image resolution also limits the impact of the work. While it may be a perfectly reasonably thing to study the problem initially at lower resolution, I believe that most people care about the models with the best metrics, and all of those methods today use image tiling to handle high resolution images.

**Questions:**

How will the results change at higher resolution and what would the comparison look like when compared against tile-based methods like LLava-1.6 or any of the more recent VLM work like Qwen2-VL etc ...?

---

> ### Author Response · Authors · 2024-11-20
>
> We thank Reviewer MCFE very much for the insightful feedback.
> We are particularly grateful for your acknowledgment of the "step forward" our technique represents.
> Your recognition of the "significant" improvements in metrics is highly encouraging.
> Point-to-point responses are provided below.
>
>
> **W1: Clean A/B Experiment with Extended Evaluation Benchmarks.**
>
>
> **A1:**
> We apologize that the provided A/B experiment in Table 3 is not comprehensive enough.
> Following your suggestion, we extend the  A/B experiment in Table 3 by incorporating more benchmarks such as TextVQA and DocVQA, providing a more balanced and representative distribution of tasks, where scores of OCRBench are divided by 10 for computing averaged scores.
>
> Empirical results in the following table demonstrate that our proposed vision-centric supervision utilized by Ross leads to significant improvements in most cases.
> Moreover, we found Ross contributes more significant improvements over fine-grained comprehension datasets, such as HallusionBench, MMVP, ChartQA, and OCRBench.
>
> | Benchmark | CLIP  | | | | SigLIP | | | |
> |-|-|-|-|-|-|-|-|-|
> | LLM |  Vicuna | | Qwen2 | | Vicuna | | Qwen2 | |
> | Method | LLaVA | Ross | LLaVA | Ross | LLaVA | Ross | LLaVA | Ross |
> | POPE-acc | 86.3 | **87.2 ↑ 0.9** | 87.9 | **88.4 ↑ 0.5** | 86.0 | **87.7 ↑ 1.7** | 88.5 | **88.7 ↑ 0.2** |
> | HallusionBench-aAcc | 52.5 | **55.8 ↑ 3.3** | 55.0 | **59.1 ↑ 4.1** | 50.4 | **53.8 ↑ 3.4** | 57.3 | **58.2 ↑ 0.9** |
> | MMBench-EN-dev | 67.0 | **67.6 ↑ 0.6** | 73.8 | **75.2 ↑ 1.4** | 64.5 | **69.2 ↑ 4.7** | 76.3 | **76.9 ↑ 0.6** |
> | MMBench-CN-dev | **60.0** | 59.8 ↓ 0.2 | 72.9 | **73.7 ↑ 0.8** | 63.1 | **63.4 ↑ 0.3** | 75.7 | **76.3 ↑ 0.7** |
> | SEED-img | **66.7** | 66.4 ↓ 0.3 | 70.3 | **70.7 ↑ 0.4** | 68.2 | **69.0 ↑ 0.8** | **72.3** | 72.1 ↓ 0.2 |
> | MMMU-dev | 30.0 | **34.0 ↑ 4.0** | 44.0 | **45.3 ↑ 1.3** | 33.3 | **38.0 ↑ 4.7** | 38.7 | **41.3 ↑ 2.6** |
> | MMMU-val | 35.3 | **36.0 ↑ 0.7** | 41.9 | **42.6 ↑ 0.7** | 34.2 | **35.4 ↑ 1.2** | 41.8 | **43.8 ↑ 2.0** |
> | MMVP | 28.0 | **36.3 ↑ 8.3** | 29.6 | **42.2 ↑ 12.6** | 27.3 | **38.0 ↑ 10.7** | 40.7 | **49.3 ↑ 8.6** |
> | AI2D-test | 61.2 | **61.4 ↑ 0.2** | 71.9 | **73.3 ↑ 1.4** | **62.6** | 62.4 ↓ 0.2 | 74.0 | **74.5 ↑ 0.5** |
> | ChartQA-test | 32.9 | **39.8 ↑ 6.9** | 36.2 | **41.6 ↑ 5.4** | 34.0 | **48.2 ↑ 14.2** | 44.4 | **46.9 ↑ 2.5** |
> | DocVQA-val | 33.4 | **41.6 ↑ 8.2** | 31.1 | **44.7 ↑ 13.6** | 40.4 | **40.7 ↑ 0.3** | 39.2 | **39.3 ↑ 0.1** |
> | InfoVQA-val | 21.2 | **26.4 ↑ 5.2** | 22.1 | **39.3 ↑ 16.2** | 22.8 | **23.3 ↑ 0.5** | 24.0 | **25.1 ↑ 1.1** |
> | TextVQA-val | 55.7 | **58.7 ↑ 3.0** | 52.0 | **54.1 ↑ 2.1** | 60.5 | **62.6 ↑ 2.1** | 56.3 | **57.5 ↑ 1.2** |
> | OCRBench | 339 | **350 ↑ 11** | 363 | **381 ↑ 18** | 354 | **365 ↑ 11** | 432 | **448 ↑ 16** |
> | RealWorldQA | 52.7 | **53.2 ↑ 0.5** | 56.7 | **57.4 ↑ 0.7** | 55.0 | **57.1 ↑ 2.1** | 57.9 | **59.1 ↑ 1.2** |
> | **Average** | 47.8 | **50.6 ↑ 2.8** | 52.1 | **56.4 ↑ 4.3** | 49.2 | **52.4 ↑ 3.2** | 55.4 | **56.9 ↑ 1.5** |
>
> CLIP: CLIP-ViT-L/14@336; SigLIP: SigLIP-SO400M-ViT-L/14@384; Vicuna: Vicuna-7B-v1.5; Qwen2: Qwen2-7B-Instruct
>
> **W2 & Q1: Comparison on High-Resolution Benchmarks.**
>
> **A2:**
> We appreciate the feedback regarding the need for high-resolution comparisons. To address this concern, we have incorporated the "anyres" technique proposed by LLaVA-v1.6 into our Ross. Specifically, for each image, we employ a grid configuration of 384×{2×2, 1×{2,3,4}, {2,3,4}×1} to identify the input resolution, resulting in a maximum of 5×729 = 3,645 visual tokens. Each 384×384 crop is required to reconstruct the original input via the denoising objective proposed by Ross. In the following table, our ROSS-7B-anyres surpasses LLaVA-v1.6-7B and Cambrian-1-8B in most cases. These results indicate that Ross not only performs well at lower resolutions but also maintains its competitive edge at higher resolutions, making it a robust and versatile method.
>
> | Model                    | ChartQA | DocVQA | InfoVQA | TextVQA | OCRBench | RealWorldQA |
> |--------------------------|---------|--------|---------|---------|----------|-------------|
> | GPT-4V-1106              | 78.5    | 88.4   | --      | 78.0    | 645      | 61.4        |
> | Gemini-1.5 Pro          | 81.3    | 86.5   | --      | 78.1    | --       | 67.5        |
> | Grok-1.5                | 76.1    | 85.6   | --      | 78.1    | --       | 68.7        |
> | |
> | LLaVA-v1.5-7B    | 18.2    | 28.1   | 25.7    | 58.2    | 317      | 54.9        |
> | LLaVA-v1.6-7B   | 65.5    | 74.4   | 37.1    | 64.8    | 532      | 57.6        |
> | Cambrian-1-8B           | 73.3    | 77.8   | --      | 71.7    | **624**  | 64.2        |
> | **Ross-7B-anyres** | **76.9** | **81.8** | **50.5** | **72.2** | 607 | **66.2** |

---

> > ### Author Response · Authors · 2024-11-27
> >
> > Dear Reviewer MCFE,
> >
> > As the ICLR discussion phase is nearing its conclusion, we are writing to kindly ask that you review our responses to the comments and questions raised during the review process. Your thorough examination and any additional feedback or discussions you may wish to initiate will be crucial in refining our work. We look forward to your final ratings and any further dialogue that may enhance our paper.
> >
> > Sincerely,
> >
> > Authors

---

> > > ### Author Response · Authors · 2024-12-02
> > >
> > > Thank you again for your insightful and positive feedback. We believe our rebuttal has addressed your questions and concerns. With the discussion phase deadline approaching, we would greatly appreciate it if you could let us know if you have any additional questions. We are happy to respond as soon as possible.

---

### Official Review · Reviewer_kvNb · 2024-11-08

**Soundness:** 3
**Presentation:** 3
**Contribution:** 3
**Rating:** 6
**Confidence:** 4

**Summary:**

The paper proposes a new LMM training approach with an additional branch for input image reconstruction. The results show the model enhances fine-grained comprehension and reduces hallucinations. ROSS employs a denoising objective to address spatial redundancy that reconstructs latent visual tokens rather than raw RGB values. Empirical evaluations demonstrate that ROSS consistently outperforms conventional LMMs using single or multiple visual encoders on visual understanding benchmarks.

**Strengths:**

**Novel Image-Based Supervision**: ROSS leverages image reconstruction as a supervisory signal, enabling the model to capture fine-grained visual features and semantics that significantly reduce hallucination artifacts compared to text-supervised approaches. The idea conceptually makes sense and is proven in the experiments.

**Comprehensive Analysis of Model Variants**: The paper provides a thorough study of various architectural choices and configurations within the ROSS framework, offering insight into optimal setups.

**Empirical Validation**: Extensive ablation studies and benchmark evaluations demonstrate ROSS's superior performance metrics, particularly in tasks requiring high-fidelity visual understanding, with statistically significant improvements over state-of-the-art baselines.

**Weaknesses:**

- Potential Unfairness in Comparisons: While the paper includes an ablation study where variables like training data are controlled for fair comparison with other models, its main results table appears to use different datasets compared to competing methods. This inconsistency in data setup might lead to an unfair advantage for ROSS, making it difficult to assess the true comparative effectiveness of the approach against state-of-the-art methods.

- Computational Overhead: The denoising process introduces extra computational overhead during training, but the paper does not quantify or discuss this cost, leaving readers uncertain about the practical trade-offs of using this approach.

- Limited Analysis of Generation vs. Reconstruction Performance: The paper compares ROSS’s reconstructive approach to generative methods, noting that the generative approach underperforms in comprehension tasks. However, it lacks a thorough exploration of why the generative method yields lower performance. A more in-depth discussion of the limitations and differences between the two approaches would enhance understanding and help identify when reconstruction might be preferable to generation in multimodal tasks.

**Questions:**

It would be cool if authors could address the concerns in weakness.

---

> ### Author Response · Authors · 2024-11-20
> **Rebuttal by Authors (Part 1/2)**
>
> We thank reviewer kvNb for the valuable time and constructive feedback.
> We are truly grateful for your use of the terms "novel" and "makes sense" to describe our work.
> Your appreciation of our comprehensive analysis and the robust empirical validation is greatly encouraging.
> Point-to-point responses are provided below.
>
>
> **W1: Potential Unfairness in Comparisons.** While the paper includes an ablation study where variables like training data are controlled for fair comparison with other models, its main results table appears to use different datasets compared to competing methods. This inconsistency in data setup might lead to an unfair advantage for Ross, making it difficult to assess the true comparative effectiveness of the approach against state-of-the-art methods.
>
>
> **A1:**
> We appreciate the reviewer's concern regarding the potential unfairness in the comparisons presented in Table 4.
> We have tried our best to conduct fair comparisons against the baseline, including the same visual encoder, base language model, pre-training and instruction tuning data, where significant improvements are observed consistently, which can be found in Table 3.
>
> We acknowledge that the comparisons in Table 4 might be perceived as unfair due to the use of different datasets.
> To mitigate this concern, we have performed an additional experiment where we compare Ross with the state-of-the-art method, LLaVA, using the exact same datasets and settings.
> Specifically, we used the CLIP-ViT-L/14@336 visual encoder and the Qwen2-7B-Instruct language model.
> Empirical results below demonstrate that *Ross consistently outperforms LLaVA under these identical conditions.*
>
> | PT   | SFT  | $\mathcal{L}_{\mathrm{LMM}}^{\mathrm{visual}}$ | POPE  | Hallu. | ChartQA | OCRBench | MMB$^{\text{EN}}$ | AI2D |
> |------|------|-------------------------------------------------|-------|--------|---------|----------|-------------------|------|
> | 558K | 737K | --                                              | 87.9  | 55.0   | 34.0    | 363      | 73.8              | 72.4 |
> | 558K | 737K | ✔                                               | **88.4 ↑ 0.5** | **59.1 ↑ 4.1** | **40.4 ↑ 6.4** | **380 ↑ 17** | **75.2 ↑ 1.4** | **73.3 ↑ 0.9** |
> | 2M   | 1.2M | --                                              | 88.5  | 53.8   | 41.2    | 388      | 76.5              | 73.9 |
> | 2M   | 1.2M  | ✔                                               | **88.9 ↑ 0.4** | **57.3 ↑ 2.5** | **43.2 ↑ 2.0** | **405 ↑ 17** | **78.0 ↑ 1.5** | **74.1 ↑ 0.2** |
>
> To be honest, unfair comparison is actually a common problem in the field of LMMs as there are many challenges in conducting fully fair comparisons.
> We have listed a series of representative components that lead to this unfairness in the following table.
> It was relatively hard for researchers to conduct a completely fair comparison against the state-of-the-art methods, not to mention that some methods may contribute at the data level, *e.g.*, ShareGPT4V.
>
> | Method               | Encoder            | Resolution | # tokens | LLM                  | Pre-train | SFT  |
> |----------------------|--------------------|------------|----------|----------------------|-----------|------|
> | LLaVA-v1.5-7B        | CLIP-L/14          | 336        |          | Vicuna-7B-v1.5       | 558K      | 665K |
> | LLaVA-v1.6-7B        | CLIP-L/14          | 5 × 336    | 5 × 576  | Vicuna-7B-v1.5       | 558K      | 665K |
> | Mini-Gemini-7B       | CLIP-L/14          | 336        | 576      | Mixtral-8x7B         | 1.2M      | 1.5M |
> |                      | + ConvNeXt-L       | 768        |          |                      |           |      |
> | Cambrian-1-8B        | CLIP-L/14          | 336        | 576      | Llama3-8B-Instruct   | 2.5M      | 7M   |
> |                      | + SigLIP-L/14      | 384        |          |                      |           |      |
> |                      | + DINOv2-L/14      | 518        |          |                      |           |      |
> |                      | + ConvNeXt-XXL     | 1024       |          |                      |           |      |
> | Ross-7B              | SigLIP-L/14        | 384        | 729      | Qwen2-7B-Instruct    | 2M        | 1.2M |

---

> > ### Author Response · Authors · 2024-11-20
> > **Rebuttal by Authors (Part 2/2)**
> >
> > **W2: Computational Overhead.** The denoising process introduces extra computational overhead during training, but the paper does not quantify or discuss this cost, leaving readers uncertain about the practical trade-offs of using this approach.
> >
> >
> > **A2:**
> > We apologize for the oversight in quantifying and discussing the computational overhead introduced by the denoising process.
> > To address this concern, we have conducted additional experiments to measure the computational costs and provide a clearer understanding of the practical trade-offs.
> >
> > Evaluations are conducted using 8 A100 GPUs with a global batch size of 128.
> > Due to the limited GPU memory, we accumulate 4 gradient steps and the batch size per GPU is 4.
> > The whole stage requires 5757 training steps.
> > GPU memories are averaged over 8 GPUs with DeepSpeed Zero 3.
> > As demonstrated in the following table, the denoising process introduces a negligible increase in training time ($\approx$10\% compared to the baseline), while the benefits outweigh the minor additional costs.
> >
> > | Vision      | Base LLM             | $\mathcal{L}_{\mathrm{LMM}}^{\mathrm{visual}}$ | Trainable Parameters | Speed (s/iter) | Time       | GPU Memory  |
> > |-|-|-|-|-|-|-|
> > | CLIP-L/336  | Qwen2-7B-Instruct    | --  | 7.63 B              | 8.31           | 13h 17min  | 45.34 G     |
> > | CLIP-L/336  | Qwen2-7B-Instruct    | ✔    | 7.68 B              | 9.02 (1.09 ×)  | 14h 25min  | 46.62 G (1.03 ×) |
> > | CLIP-L/336  | Vicuna-13B-v1.5      | --    | 13.05 B             | 13.33          | 21h 19min  | 48.62 G     |
> > | CLIP-L/336  | Vicuna-13B-v1.5      | ✔                                             | 13.11 B             | 14.69 (1.10 ×) | 23h 30min  | 49.07 G (1.01 ×) |
> > | SigLIP-L/384| Qwen2-7B-Instruct    | --                                            | 7.63 B              | 8.77           | 14h 1min   | 47.08 G     |
> > | SigLIP-L/384| Qwen2-7B-Instruct    | ✔                                             | 7.68 B              | 9.48 (1.08 ×)  | 15h 9min   | 52.07 G (1.11 ×) |
> > | SigLIP-L/384| Vicuna-13B-v1.5      | --                                            | 13.05 B             | 14.22          | 22h 44min  | 48.80 G     |
> > | SigLIP-L/384| Vicuna-13B-v1.5      | ✔                                             | 13.11 B             | 15.32 (1.08 ×) | 24h 30min  | 52.68 G (1.08 ×)
> >
> >
> > **W3: Limited Analysis of Generation vs. Reconstruction Performance.** The paper compares Ross’s reconstructive approach to generative methods, noting that the generative approach underperforms in comprehension tasks. However, it lacks a thorough exploration of why the generative method yields lower performance. A more in-depth discussion of the limitations and differences between the two approaches would enhance understanding and help identify when reconstruction might be preferable to generation in multimodal tasks.
> >
> >
> > **A3:**
> > First, we would like to clarify that most generative methods, such as Chameleon and Show-o, aim to equip both comprehension and creation within a *single* model, which often *underperforms in comprehension tasks.*
> > We hypothesize that the underlying reason for the lower performance of generative methods in comprehension tasks is **the weak correspondence between inputs and supervision** under generative settings, which typically arises from both the (1) data and the (2) design of these methods.
> >
> > (1) Typical generative methods that explore the synergy of comprehension and generation, usually leverage image generation conditioned on text instructions on *(i) text-to-image datasets* or *(ii) interleaved datasets* as extra supervision.
> > However, (i) text-to-image datasets are typically designed to generate *high-aesthetic* samples rather than text-aligned ones, and (ii) interleaved datasets aim to enable few-shot learning via interleaving independent supervised examples, where reasoning becomes more important than alignment.
> > Therefore, there exists a clear disconnect where the supervision (image) has little to do with the input (text instruction).
> > For example, the CLIP-Score, which measures the similarity between text and images, is only 0.3043 for the LAION-Art dataset and 0.2842 for the MMC4 dataset, indicating that the supervision signals in these datasets are \textit{not} well-suited for tasks requiring strong text-image alignment.
> >
> > (2) Even when we attempt to ensure image-text alignment by converting aligned caption data into creation data for supervision, the results demonstrated in Table 2 remain unsatisfactory.
> > This suggests that the *design of generative objectives itself does not inherently require a strong correspondence* between inputs and supervision targets.
> >
> > In contrast, reconstructive methods like Ross leverage the original input images as auxiliary supervision, ensuring a strong and direct correspondence, which is crucial for tasks requiring accurate comprehension and interpretation of multimodal data, leading to significantly improved performance.

---

> > > ### Author Response · Authors · 2024-11-27
> > >
> > > Dear Reviewer kvNb,
> > >
> > > As the ICLR discussion phase is nearing its conclusion, we are writing to kindly ask that you review our responses to the comments and questions raised during the review process. Your thorough examination and any additional feedback or discussions you may wish to initiate will be crucial in refining our work. We look forward to your final ratings and any further dialogue that may enhance our paper.
> > >
> > > Sincerely,
> > >
> > > Authors

---

> > > > ### Comment · Reviewer_kvNb · 2024-11-29
> > > >
> > > > My concern is addressed. Thanks!

---

> > > > > ### Author Response · Authors · 2024-11-30
> > > > >
> > > > > Thank you for your valuable review. Your insights greatly improve our work. If any of your concerns have been addressed, could you please consider increasing the score?

---

### Author Response · Authors · 2024-11-20
**Summary of Revisions**

We sincerely thank all reviewers for their valuable and constructive comments. We have tried our
best to revise the paper to address all concerns. **All revisions are marked in purple.** Specifically:

- **@Reviewer kvNB**, **Reviewer SsLG**, and **Reviewer rXNr,** we have added discussions on
computational costs in Section 5.2 and detailed comparisons at Table 10 in Appendix B,
where our Ross brings marginal computational overhead.
- **@Reviewer SsLG**, to better illustrate our insight, we have provided reconstruction results
in Figure 9 in Section 5.2, where high-level features of Ross-7B can be projected back into
the pixel space.
- **@Reviewer SsLG**, we have provided an ablation on the schedule of β in Table 11 in
Appendix C.1, where our Ross is robust against different denoising schedules.
- **@Reviewer MCFE** and **Reviewer rXNr**, we have provided a more comprehensive A/B
experiment in Table 12 in Appendix C.1, where the proposed vision-centric objective brings
significant improvements in most cases.
- **@Reviewer MCFE**, we have incorporated the “anyres” technique and compared state-of-
the-art alternatives on high-resolution benchmarks in Table 13 in Appendix C.2, where our
Ross-7B-anyres surpasses LLaVA-v1.6-7B and Cambrian-1-8B under most cases.
- **@Reviewer tGMa**, we have studied the impact of Ross on language capabilities in Table 14
in Appendix C.3, where Ross does not harm language capabilities as it brings improvements
in most cases.
- **@Reviewer tGMa**, we have investigated the effectiveness of Ross across different model
sizes in Table 15 in Appendix C.3, where Ross brings improvements over the baseline
(LLaVA) across different model sizes in most cases.
- **@Reviewer kvNb** and **Reviewer tGMa**, we have studied the data scaling property of our
Ross in Table 16 in Appendix C.3, where Ross consistently brings significant improvements
as the training data scale increases.
- **@Reviewer tGMa**, to better explain the reasoning behind how the vison-centric supervision
enables the model to focus on relevant areas, we visualize the gradient using GradCAM in
Figure 12 in Appendix C.3, where Ross generates reasonable gradients as training evolves,
guiding the model to focus on the relevant regions of the image.

Please let us know if you have any further questions. We are always looking forward to open
discussions.

Sincerely,

Authors

---

### Meta-Review · Area_Chair_vqos · 2024-12-19

**Metareview:**

This paper introduces a new approach to visual instruction tuning by incorporating an additional reconstruction objective. The proposed method is interesting and demonstrates clear improvements. Reviewers provided overall positive feedback, and the authors submitted strong rebuttals addressing their comments. Taking all reviewers' feedback into consideration, the AC has recommended the paper for acceptance.

**Additional Comments On Reviewer Discussion:**

The overall feedback from the final reviews is positive. The AC disagrees with reviewer rXNr's comments regarding novelty--using existing techniques from other areas does not mean the method lacks originality. While the concern about efficiency is acknowledged, it is deemed secondary given the new insights provided by the paper. The authors are encouraged to incorporate the feedback from the reviewers to improve their manuscript.

---

### Decision · Program_Chairs · 2025-01-22

Accept (Poster)